# Low-Rank Subspaces in GANs

**Jiapeng Zhu**[1]  **Ruili Feng**[2,3]  **Yujun Shen**[4]  **Deli Zhao**[2]
**Zheng-Jun Zha**[3]  **Jingren Zhou**[2]  **Qifeng Chen**[1]*
[1]Hong Kong University of Science and Technology  [2]Alibaba Group
[3]University of Science and Technology of China  [4]ByteDance Inc.
{jengzhu0, ruilifengustc, shenyujun0302, zhaodeli}@gmail.com
zhazj@ustc.edu.cn  jingren.zhou@alibaba-inc.com  cqf@ust.hk

## Abstract

The latent space of a Generative Adversarial Network (GAN) has been shown to encode rich semantics within some subspaces. To identify these subspaces, researchers typically analyze the statistical information from a collection of synthesized data, and the identified subspaces tend to control image attributes globally (*i.e.*, manipulating an attribute causes the change of an entire image). By contrast, this work introduces *low-rank subspaces* that enable more precise control of GAN generation. Concretely, given an arbitrary image and a region of interest (*e.g.*, eyes of face images), we manage to relate the latent space to the image region with the Jacobian matrix and then use low-rank factorization to discover steerable latent subspaces. There are three distinguishable strengths of our approach that can be aptly called LowRankGAN. First, compared to analytic algorithms in prior work, our low-rank factorization of Jacobians is able to find the low-dimensional representation of attribute manifold, making image editing more precise and controllable. Second, low-rank factorization naturally yields a *null space* of attributes such that moving the latent code within it only affects the outer region of interest. Therefore, *local image editing* can be simply achieved by projecting an attribute vector into the null space without relying on a spatial mask as existing methods do. Third, our method can robustly work with a local region from one image for analysis yet well generalize to other images, making it much easy to use in practice. Extensive experiments on state-of-the-art GAN models (including StyleGAN2 and BigGAN) trained on various datasets demonstrate the effectiveness of our LowRankGAN[1].

## 1 Introduction

Generative Adversarial Networks (GANs), which can produce high-fidelity images visually indistinguishable from real ones [18, 19, 4], have made tremendous progress in many real-world applications [16, 39, 32, 10, 38, 27, 7, 34]. Some recent studies [12, 26, 17] show that pre-trained GAN models spontaneously learn rich knowledge in latent spaces such that moving latent codes towards some certain directions can cause corresponding attribute change in images. In this way, we are able to control the generation process by identifying semantically meaningful latent subspaces.

For latent semantic discovery, one of the most straightforward ways is to first generate a collection of image synthesis, then label these images regarding a target attribute, and finally find the latent separation boundary through supervised training [12, 26, 17]. Prior work [30, 25, 14] has pointed out that the above pipeline could be limited by the labeling step (*e.g.*, unable to identify semantics

---

*denotes the corresponding author.
[1]Code is available at https://github.com/zhujiapeng/LowRankGAN/.

beyond well-defined annotations) and proposed to find steerable directions of the latent space in an unsupervised manner, such as using Principal Component Analysis (PCA) [14]. However, existing approaches prefer to discover global semantics, which alter the entire image as a whole and fail to relate a latent subspace to some particular image region. Consequently, they highly rely on spatial masks [29, 8, 2] to locally control the GAN generation, which is of more practical usage.

In this work, we bridge this gap by introducing a *low-rank subspace* of GAN latent space. Our approach, called *LowRankGAN*, starts with an arbitrary synthesis together with a region of interest, such as the eyes and nose in a face image or the ground in a scene image. Then, we manage to relate the latent space and the image region via computing the Jacobian matrix with the pre-trained generator. After that comes the core part of our algorithm, which is to perform low-rank factorization based on the resulting Jacobian matrix. Different from other analytic methods, the low-rank factorization of Jacobians is capable of uncovering the intrinsic low-dimensional representation of attribute manifold, thus helping solve more precise editing direction to the target attribute. Besides, the low-rank factorization naturally yields a *null space* of attributes such that moving the latent code within this subspace mainly affects the remaining image region. With such an appealing property, we can achieve *local image control* by projecting an attribute vector into the null space and using the projected result to modulate the latent code. This observation is far beyond trivial because the latent code is commonly regarded to act on the entire feature map of GAN generator [13, 4, 18]. We also find that the low-rank subspace derived from local regions of one image is widely applicable to other images. In other words, our approach *requires only one sample* instead of abundant data for analysis. More astonishingly, supposing that we identify a changing-color boundary on one car image, we can use this boundary to change the color of cars from other images, even the cars locate at a different spatial area or with different shapes. This suggests that the GAN latent space is locally semantic-aware. Moreover, LowRankGAN is strongly robust to the starting image region for Jacobian matrix computation. For instance, to control the sky in synthesized images, our algorithm does not necessarily rely on a rigid sky segmentation mask but can also satisfyingly work with a casual bounding box that marks the sky region (partially or completely), making it much easy to use in practice.

Our main contributions are summarized as follows. First, we study low-rank subspaces of the GAN latent space and manage to locally control the GAN image generation by simply modulating the latent code without the need for spatial masks. Second, our method can discover latent directions regarding local regions of only one image yet generalize to all images, shedding light on the intrinsic semantic-aware structure of GAN latent space. Third, we conduct comprehensive experiments to demonstrate the effectiveness and robustness of our algorithm on a wide range of models and datasets.

## 2   Related Work

Interpreting well-trained GAN models [1, 17, 26] has recently received wide attention because this can help us understand the internal representation learned by GANs [34] and further control the generation process [35, 38]. It is shown that some subspaces in the GAN latent space are highly related to the attributes occurring in the output images. To identify these semantically meaningful subspaces, existing approaches fall into two folds, *i.e.*, supervised [12, 27, 17, 22] and unsupervised [30, 14, 25, 28]. On the one hand, supervised approaches seek help from off-the-shelf attribute classifiers [12, 27] or simple image transformations [17, 22] to annotate a set of synthesized data and then learn the latent subspaces from the labeled data. On the other hand, unsupervised approaches try to discover steerable latent dimensions by statistically analyzing the latent space [14], maximizing the mutual information between the latent space and the image space [30], or exploring model parameters [25, 28]. However, all of the subspaces detected by these approaches tend to control global image attributes such that the entire image will be modified if we edit a particular attribute. To achieve local image control, a common practice is to introduce a spatial mask [29, 8, 2, 34, 37], which works together with the intermediate feature maps of the GAN generator.

Compared to prior work, our LowRankGAN has the following differences: no spatial masks of objects are needed to perform precise local editing, and generic attribute vectors can be obtained with only one image synthesis and arbitrary local regions of interest without any training or statistical analysis on a large amount of data. To our best knowledge, we are the first to demonstrate the effectiveness of low-rank factorization in interpreting generative adversarial models. Although there are also some works using the Jacobian matrix to analyze the GAN latent space [24, 6, 31], the low-rank subspace enables precise generation control by providing the indispensable null space.

# 3 Low-Rank GAN

Let $z \in \mathbb{R}^{d_z}$ and $G(\cdot)$ denote the input latent code and the generator of a GAN, respectively. Prior work [12, 26, 17] has observed the relationship between some latent subspaces and image attributes such that we can achieve semantic manipulation on the synthesized sample, $G(z)$, by linearly transforming the latent code

$$x^{\text{edit}} = G(z + \alpha n), \tag{1}$$

where $n$ denotes an attribute vector within the latent space, and $\alpha$ is the editing strength. However, the latent subspaces identified by existing approaches tend to manipulate the entire image as a whole and fail to control a local image region. Differently, our proposed LowRankGAN confirms that Eq. (1) can also be used to locally control the image generation. The following parts describe how to discover the low-rank subspaces from a pre-trained GAN model as well as its practical usage.

## 3.1 Degenerate Jacobian

First, we give the definition of Jacobian matrix. Let the real image $x \in \mathcal{R}^{d_x}$ be in the $d_x$-dimensional space and the input prior $z \in \mathcal{R}^{d_z}$ be of dimension $d_z$. For an arbitrary point $z$, the Jacobian matrix $J_z$ of the generator $G(\cdot)$ with respect to $z$ is defined as $(J_z)_{j,k} = \frac{\partial G(z)_j}{\partial z_k}$, where $(J_z)_{j,k}$ is the $(j,k)$-th entry of $J_z \in \mathcal{R}^{d_x \times d_z}$. By the Taylor series, the first-order approximation to the edited result can be written as

$$G(z + \alpha n) = G(z) + \alpha J_z n + o(\alpha). \tag{2}$$

An effective editing direction should significantly alter $G(z)$, which can be attained by maximizing the following variance

$$n = \underset{\|n\|_2=1}{\arg\max} \|G(z + \alpha n) - G(z)\|_2^2 \approx \underset{\|n\|_2=1}{\arg\max} \alpha^2 n^T J_z^T J_z n. \tag{3}$$

It is easy to know that the above optimization admits a closed-form solution, which is the eigenvector associated with the largest eigenvalue. Therefore, the subspace spanned by editing directions is the principal subspace of $J_z^T J_z$.

For GAN architectures, previous work [11] has proven and demonstrated that the Jacobian matrix $J_z^T J_z$ is degenerate, which is described by the following theorem.

**Theorem 1** *Assume that the real data $\mathcal{X}$ lie in some underlying manifold of dimension $dim(\mathcal{X})$ embedded in the pixel space. Let $z_k \in \mathcal{R}^{d_z}$ denote the latent code of the k-th layer of the generator of GAN over $\mathcal{X}$ and $J_{z_k}$ be the Jacobian of the generator with respect to $z_k$. Then we have $rank(J_{z_{k+1}}^T J_{z_{k+1}}) \leq rank(J_{z_k}^T J_{z_k}) \leq d_z$, where $rank(J_z^T J_z)$ denotes the rank of the Jacobian. Further, if $dim(\mathcal{X}) < d_z$, then $rank(J_{z_k}^T J_{z_k}) < d_z$.*

Theorem 1 says that the rank of the Jacobian $J_z^T J_z$ might be progressively reduced, and the manifold characterized by the generator admits a low-dimensional structure under a mild condition. Therefore, the attribute subspace must admit the analogous structure, which can be found by low-rank factorization of the Jacobian $J_z^T J_z$.

## 3.2 Low-Rank Jacobian Subspace

In practice, the numerical evaluation indicates that the Jacobian is actually of full rank, implying that it is perturbed with noise during the highly nonlinear generation process. If we assume that such perturbation is sparse, then the low-dimensional structure of the attribute manifold can be perfectly revealed by low-rank factorization of $J_z^T J_z$ with corruption.

Given a low-rank matrix $M$ corrupted with sparse noise, the low-rank factorization can be formulated as $M = L + S$, where $L$ is the low-rank matrix and $S$ is the corrupted sparse matrix. Directly solving this problem is actually very hard due to the non-convexity of the optimization. Hopefully, it can reduce to a tractable convex problem [33]:

$$\min_{L,S} \|L\|_* + \lambda \|S\|_1 \qquad \text{s.t.} \quad M = L + S, \tag{4}$$

where $\|L\|_* = \sum_i \sigma_i(M)$ is the nuclear norm defined by the sum of all singular values, $\|S\|_1 = \sum_{ij} |M_{ij}|$ is the $\ell_1$ norm defined by the sum of all absolute values in $S$, and $\lambda$ is a positive weight

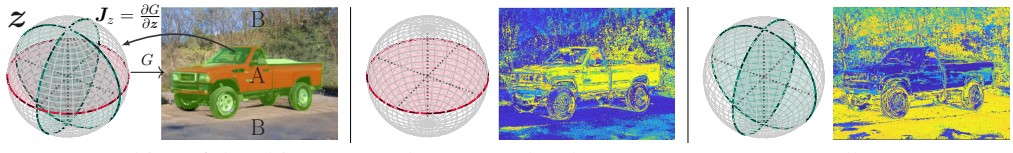

| (a) Jacobian of the object | (b) Low-rank subspace | (c) Null space |

Figure 1: **Framework of LowRankGAN**, which is able to precisely control the image generation of GANs. (a) Given an arbitrary synthesis and a region of interest, *i.e.*, region A under **mask**, we relate the image region to the latent space with Jacobian matrix. The proposed low-rank subspace (**red** circle) is derived by performing low-rank factorization on the Jacobian, which naturally yields a null space (**green** circles). (b, c) The low-rank subspace and the null space correspond to the editing of region A and region B, respectively.

parameter to balance the sparsity and the low rank. The convex optimization in Eq. (4) is also referred to as *Principal Component Pursuit (PCP)* [5], which can be solved by the Alternating Directions Method of Multipliers (ADMM) [21, 3]. The detailed introduction of low-rank factorization and ADMM is presented in the *Supplementary Material*. Suppose that the solution for $\boldsymbol{J}_z^T \boldsymbol{J}_z$ in Eq. (4) is

$$\boldsymbol{M} = \boldsymbol{J}_z^T \boldsymbol{J}_z = \boldsymbol{L}^* + \boldsymbol{S}^*, \tag{5}$$

where $\boldsymbol{L}^*$ is the low-rank representation of $\boldsymbol{J}_z^T \boldsymbol{J}_z$, which has rank $r$ corresponding to $r$ attributes in images. By singular value decomposition (SVD), we are able to specify those attributes

$$\boldsymbol{U}, \boldsymbol{\Sigma}, \boldsymbol{V}^T = \text{SVD}(\boldsymbol{L}^*), \tag{6}$$

where $\boldsymbol{\Sigma}$ is a diagonal matrix sorted by singular values, $\boldsymbol{U}$ and $\boldsymbol{V} = [\boldsymbol{v}_1, \dots, \boldsymbol{v}_r, \dots, \boldsymbol{v}_{d_z}]$ are the matrices of left and right singular matrices containing $r$ attribute vectors associated with $r$ attributes. With $\boldsymbol{V}$, we can simply edit a specific attribute of an image by using $[\boldsymbol{v}_1, \dots, \boldsymbol{v}_r]$ according to Eq. (1). In this paper, we mainly focus on the local control of an image. So we compute the Jacobian matrix with a specified region $\boldsymbol{M}_{\text{region}}$ in an image.

### 3.3   Precise Control via Null Space Projection

Here, a precise control means that the pixels in the remaining region change as little as possible when editing a specific region. For instance, if we want to close the eyes in a face image, the ideal scenario is that the other regions, such as mouth, nose, hair, and background, will keep unchanged. Previous works [14, 25, 22, 30] could edit some attributes in a specific region but without any control on the rest region, which actually results in a global change on the image when editing the region of interest. We solve the problem via algebraic principle and propose a universal method to fulfill the precise control by utilizing the null space of the Jacobian.

As shown in Fig. 1, if we want to edit some attributes (e.g., change color) of region A (the car body in the image), the ideal manipulation is that the remaining region B in the image will preserve unedited. A natural idea is that we could project the specific attribute vector $\boldsymbol{v}_i$ of region A into a space where the perturbation on the attribute direction has no effect on region B yet has an influence on region A. Specifically, we have attained $r_a$ attribute vectors in $\boldsymbol{V}$ that can change the attributes of region A, and the rest $d_z - r_a$ vectors in $\boldsymbol{V}$ barely influence the region A. Similarly, we can also attain the attribute matrix $\boldsymbol{B} = [\boldsymbol{b}_1, \dots, \boldsymbol{b}_{d_z}]$ for region B via Eq. (6), which has $r_b$ attribute vectors changing region B, and the remaining $d_z - r_b$ vectors barely influence the region B but contains some vectors that have some influence on region A. Note that $r_a, r_b$ are the rank of the matrices on regions A and B after performing low-rank factorization, respectively. We let $\boldsymbol{B} = [\boldsymbol{B}_1, \boldsymbol{B}_2]$, where $\boldsymbol{B}_1 = [\boldsymbol{b}_1, \dots, \boldsymbol{b}_{r_b}]$ and $\boldsymbol{B}_2 = [\boldsymbol{b}_{r_b+1}, \dots, \boldsymbol{b}_{d_z}]$. Thus, the change in region B can be relieved by projecting the attribute vector $\boldsymbol{v}_i$ of region A to the orthogonal complementary space of $\boldsymbol{B}_1$, which we formulate as

$$\boldsymbol{p} = (\boldsymbol{I} - \boldsymbol{B}_1 \boldsymbol{B}_1^T)\boldsymbol{v}_i = \boldsymbol{B}_2 \boldsymbol{B}_2^T \boldsymbol{v}_i, \tag{7}$$

where $\boldsymbol{I}$ is the identity matrix. For our low-rank case, it is easy to know that $\boldsymbol{B}_2$ is the null space of the Jacobian of region B, i.e., the subspace associated with zero singular values.

However, a complete precise control might restrict the degree of editing in some special circumstances. For instance, the size of eyes will be limited if the eyebrows are kept still during editing. Or the edition of opening the mouth without any change of the chin is nearly impossible. Considering this factor, we

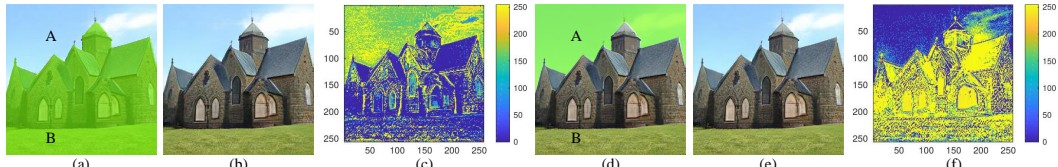

Figure 2: **Illustration of the effect of null spaces**. (a, d) The **masked** regions B and A are used to obtain the null spaces through the low-rank decomposition of the Jacobians. (b, e) The images are manipulated by altering the latent code within the derived null spaces. (c, f) The heatmaps of the $\ell_1$ loss between (a) and (b), (d) and (e), respectively, suggesting that editing within the null space barely affects the masked region of interest.

propose *precision relaxation* to alleviate this issue. Recall that the total precise control is performed by projecting $v_i$ into the null space of $B$, where all the singular values are zeros. Hence, we can involve several attribute vectors that have small singular values in order to increase the diversity. We define the $r_{\text{relax}}$ as the number of vectors used for relaxation. So the projection subspace becomes $B_1 = [b_1, \ldots, b_{r_{\text{relax}}}]$ and $B_2 = [b_{r_{\text{relax}}}, \ldots, b_{r_b+1}, \ldots, b_{d_z}]$. Obviously, the larger $r_{\text{relax}}$ is, the larger diversity we obtain. Otherwise, a small $r_{\text{relax}}$ will result in little diversity.

## 4 Experiments

In this section, we conduct experiments on two state-of-the-art GAN models, StyleGAN2 [19] and BigGAN [4], to validate the effectiveness of our proposed method. The datasets we use are diverse, including FFHQ [18], LSUN [36] and ImageNet [9]. For metrics, we use Fréchet Inception Distance (FID) [15], Sliced Wasserstein Distance (SWD) [23], and masked Mean Squared Error (MSE). For segmentation models, [20] is used for face data, and PixelLib is used for LSUN. First, we show the property of the null space obtained by low-rank decomposition and then show the improvement brought by the null space projection. Second, the direction we obtain from one image could be easily applied to the other images are verified. Third, the strong robustness to the mask is tested as well, and several state-of-the-art methods are chosen for comparison. At last, the ablation study on the parameter $\lambda$ in Eq. (4) and parameter $r_{\text{relax}}$ in Sec. 3.3 can be found in the *Supplementary Material*.

### 4.1 Precise Generation Control with LowRankGAN

**Property of Null Spaces.** First, we show the property of the null space obtained by the low-rank decomposition using a church model. For convenience, we separate the church image into two parts, e.g., the sky and the rest, as shown in Fig. 2a and Fig. 2d. According to Sec. 3.2, we obtain $J_z^T J_z = M_B$ for the masked region in Fig. 2a. And then, the low-rank decomposition on $M_B$ is conducted to get the null space of region B. We choose the vector in the null space to edit Fig. 2a and get the editing result as shown in Fig. 2b. Fig. 2c shows the $\ell_1$ loss heatmap between Fig. 2a and Fig. 2b, in which we could find the null space of region B will affect region A yet could keep region B nearly unchanged. All pixel values of Fig. 2a and Fig. 2b are in range $[0, 255]$ when computing the $\ell_1$ loss. The same process is conducted on Fig. 2d-f, in which we could also find that the null space of region A has an influence on region B but has little influence on region A. Hence, we could project the principal vectors of region A into the null space of region B to edit region A yet keep region B unchanged, thus accomplishing the precise generation control only by linear operation of latent codes.

**Editing with Null Space Projection.** Here, we will validate the effectiveness of the null space projection proposed in Sec. 3.3 on StyleGAN2 and BigGAN. For convenience, let $v_i$ represent the attribute vector obtained by Eq. (6) of a specific region and $p_i$ denote the projected vector through Eq. (7) into the null space. First, we study via StyleGAN2 pre-trained on FFHQ [18]. The eyes and the mouth are chosen as the regions to edit. As shown in Fig. 3, the remaining region changes violently when only using $v_i$ to control the specific attribute. For example, when closing the eyes of the man, the other attributes such as nose, mouth, and hair have an obvious alteration, and some artifacts appear as well. And when closing the mouth, the size of the face is deformed, and the hair is increased as well. On the contrary, when using $p_i$ to control the attributes in those images, the editing results are significantly improved, and the changes in the other regions are barely perceptible by human eyes. Second, we study via BigGAN on two categories. As shown in Fig. 3, we select the object center as

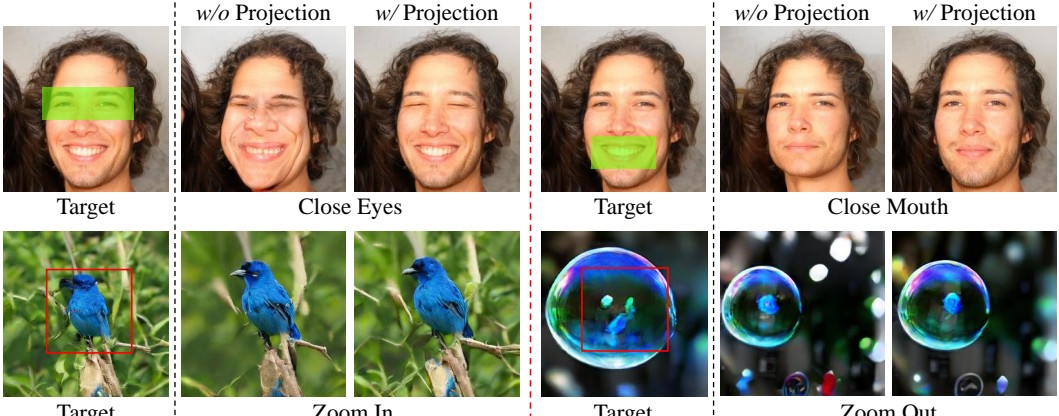

| | *w/o* Projection | *w/* Projection | | *w/o* Projection | *w/* Projection |

Target — Close Eyes — Target — Close Mouth

Target — Zoom In — Target — Zoom Out

Figure 3: **Precise generation control via the null space projection** on StyleGAN2 [19] and BigGAN [4]. For each image attribute, the original attribute vector identified from the region of interest (under **green** masks or **red** boxes) acts as changing the image globally (*e.g.*, the face shape change in the top row and the background change in the bottom row). By contrast, our approach with null space projection leads to more satisfying local editing results.

the region of interest and then obtain the attribute vectors through $M_{\text{center}}$. Directly using the attribute vector $v_i$ will significantly change the background when editing. For instance, the background of the indigo bird will be blurred, and there appear some white dots in the background of the bubble when changing the size. Rather, when using the projected singular vector $p_i$, the surroundings can keep intact as much as possible. All these results are consistent with the phenomenon in Fig. 2.

We also give the quantitative results in Tab. 1a on StyleGAN2. Besides the FID metric, the masked Mean Squared Error (MSE) is also used to qualify the change in the rest region when editing a specific region. Taking closing eyes as an example, we do not compute the loss using the eyes and their surroundings, shown with green masks in Fig. 3. This metric measures the change in the other region when editing a specific region. Obviously, the low masked MSE means high precision control. As shown in Tab. 1a, after the null space projection, both FID and MSE metrics decrease. Both the qualitative and quantitative results show that after the null space projection, we could achieve more precise control over a specific region on StyleGAN2 and BigGAN. More results can be found in the *Supplementary Material*.

## 4.2 Generalization from One Image to Others

**Results on Versatile GANs.** A question here is whether each image needs to compute the corresponding Jacobian for editing since our method requires to compute the Jacobian of the images. The answer is no, and we show that the attribute vectors obtained by our algorithm in one image could be used to effectively edit itself or the other images. As shown in Fig. 4, the reference images and the corresponding green masks indicate the images and the regions we use to find the attributes vectors, which are then used to edit the target images. From Fig. 4, we could summarize the following advantages of our method: First, the masks are unnecessary to be aligned between the reference images and target images to manipulate the targets. For example, we obtain an attribute vector that could change the hair color from the masked region of the reference image, which can change the hair color of the target images regardless of their hairstyles. And for the car, the body of the target cars varies from the shape, pose, color, etc. However, the attribute vector attained from the masked region of the reference image successfully changed the colors of the car bodies of diverse styles. Second, for conditional GANs like BigGAN, the direction attained from one category can not only be used to edit the category itself but also be used to edit other categories. For instance, we obtain a direction from the snowbird that could change its pose, which can change the pose of the other categories such as robin, owl, dog, and butterfly. Last but not least, we achieve all the local editing by just linearly shifting the latent codes instead of operating in the feature maps, making LowRankGAN easy to use.

**Applications.** Besides the qualitative results shown in Fig. 4, we also give the quantitative results in Fig. 5. And to quantitatively measure the degree of the closed eyes or mouth, we count the pixels in the eyes or mouth using the segmentation model released by [20]. We first select 10,000 images

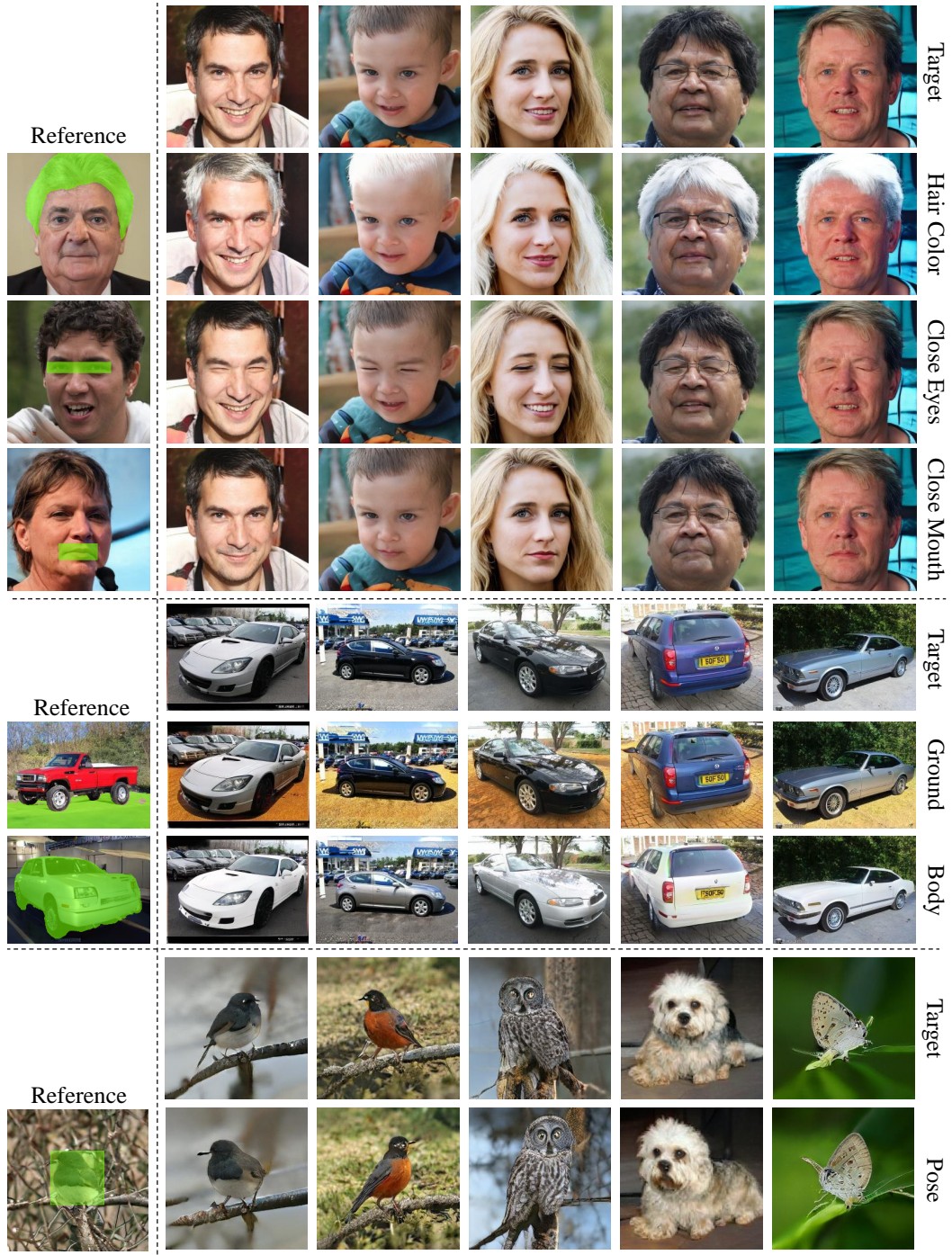

Figure 4: **Generalization of the latent semantic from the local region of one synthesis to other samples** on StyleGAN2 [19] faces, StyleGAN2 cars, and BigGAN [4]. Our approach does not require the target image to have the same masked region as the reference image. For example, cars can have different shapes and locations. The results on BigGAN even suggest that the pose semantic identified from one category can be applied to other categories.

that have open eyes when they have pixels of more than 200. After collecting images, we use the attribute vector (closing eyes) obtained from the image shown in Fig. 4 to edit those images. After editing, the number of pixels in the eyes of each image is reported in Fig. 5a. For the closed mouth, the process is exactly the same as the closed eyes, and the result is reported in Fig. 5b. We also perform statistics on the number of pixels in the eyes and mouth of the real data and synthesized data and report them in Fig. 5. Interestingly, in real data or synthesized data, only a few images have few

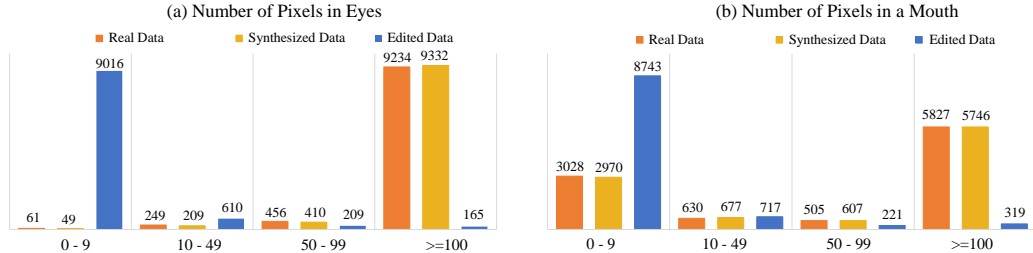

Figure 5: **Comparison among the real data distribution, synthesized data distribution, and the shifted data distribution with our LowRankGAN**, regarding closing-eyes and closing-mouth attributes. We use the number of pixels from a segmented object (*i.e.*, eyes or a mouth) to evaluate whether the eyes and mouth are closed. By analyzing *only one image*, we can make around 90% people close their eyes or mouths, yielding an application to augment the real data with rare samples.

Table 1: **Quantitative comparison results** on (a) the effect of using null space projection, and (b) the image quality from various image editing approaches [14, 25].

(a) Comparison on whether to use projection.

| Methods | Close Eyes | | Close Mouth | |
|---|---|---|---|---|
| | **FID↓** | **MSE↓** | **FID↓** | **MSE↓** |
| *w/o* Projection | 9.14 | 0.0014 | 9.47 | 0.00063 |
| *w/* Projection | **8.01** | **0.00099** | **7.69** | **0.00020** |

(b) Comparison with other methods.

| Methods | FID↓ | SWD↓ | User Study |
|---|---|---|---|
| GANSpace [14] | 7.91 | 10.46 | 5.4% |
| SeFa [25] | 7.57 | 7.50 | 37.6% |
| LowRankGAN (Ours) | **7.30** | **6.70** | **56.0%** |

pixels in the eyes (61 and 49 for the real and synthesized data, respectively). This means that the images with closed eyes are rather scarce in real or synthesized data. Thereby, a potential application of our algorithm could be data augmentation. A detailed data augmentation experiment can be found in the *Supplementary Material*.

### 4.3 Robustness of LowRankGAN to Regions of Interest

What surprises us is that our method is extremely robust to the mask (or to the region), which further enables our method could be easily used even if we do not have a segmentation model contrary to the previous works [29, 2, 34, 37]. As shown in Fig. 6, for each group, the mask varies from the size, position, and shape, but the attribute vectors we solve from those masked regions play nearly the same role when editing the target image. For instance, if we want to find some eye-related semantic, like closing eyes in Fig. 6, we could either use a segmentation model [20] to get the precise pixels of eyes or just use some coarse masks that users specify as the green masks shown in Fig. 6. The attribute vector of closing the eyes of the target can be derived from any Jacobian of those masked regions. For the church and car, we could either use the segmentation masks or the random masks as the green regions shown in Fig. 6. The same editing results can be achieved by any attribute vectors from the Jacobians of those masked regions, adding clouds to the sky of the church or changing the color of the car to the target image. One of the possible reasons for this robustness might be that the semantic in a specific region will lie in the same latent subspace. Therefore, the subspace for a sub-region of this specific region could coincide with the global one, and moving in this subspace will affect the whole region. Note that in the experiments of car and church, for the small mask regions, a relatively larger $r_{\text{relax}}$ is needed. Here we set $r_{\text{relax}} = 20$.

### 4.4 Comparison with Existing Alternatives

Now we compare our method with the state-of-the-art algorithms. We compare our method with GANSpace [14] and SeFa [25] on StyleGAN2, which are the two state-of-the-art unsupervised approaches to image editing. We find the most relevant vectors that can control the smile and hair in GANSpace and SeFa, according to their papers. As shown in Fig. 7, both GANSpace and SeFa have some degree of influence on the other region when editing a specific region. For example, when adding the smile, the identity of GANSpace is changed as well as hairstyle. When changing the hair color, GANSpace just has little activation on this attribute, while SeFa has an obvious change in the background. Meanwhile, the eyes of the women are altered as well. Instead, our method has a negligible change on other attributes when adding the smile or changing the hair color.

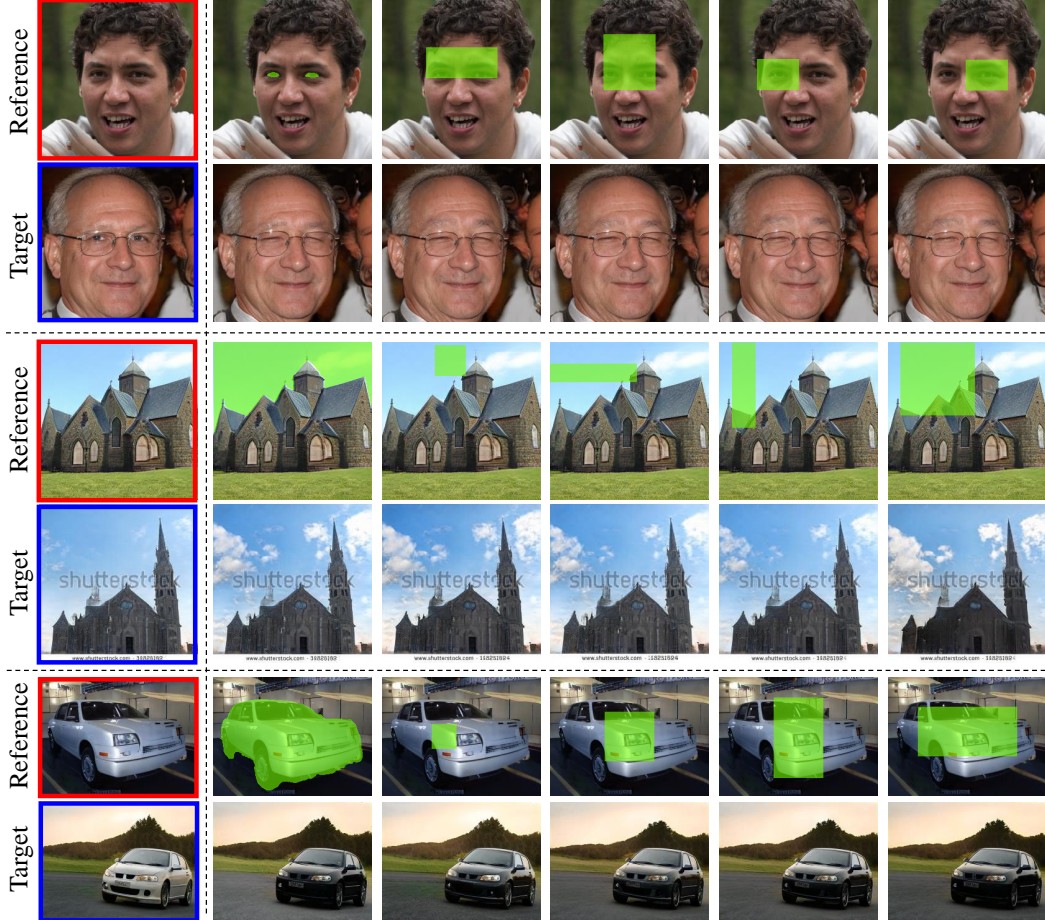

Figure 6: **Robustness of LowRankGAN to the region of interest for analysis**. For example, to edit the color of a car, our algorithm does not necessarily require a rigid segmentation mask but can also work well with an offhand bounding box and give similar results.

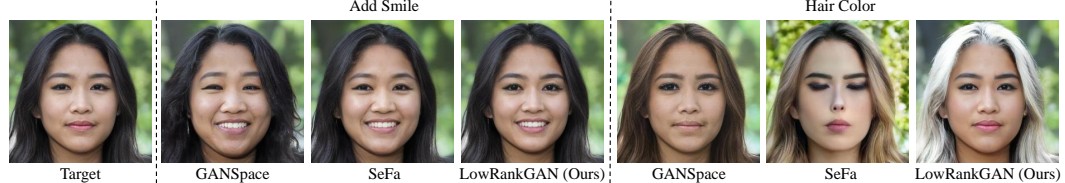

Figure 7: **Comparison on image local editing** with GANSpace [14] and SeFa [25]. Our LowRankGAN can better preserve the information beyond the target region.

Besides the qualitative results shown in Fig. 7, we also give the quantitative results in Tab. 1b on smiling. About user study, twelve students are invited to perform the evaluation. All students have a computer vision background. Each one is assigned fifty original images and the associated editing results (adding smile) obtained from different methods. Then they are asked to discriminate the image-editing quality of different methods. They are $12 \times 50 = 600$ votes. We get 595 valid votes because, in some cases, some students cannot tell which method performs best. The result of the user study is reported in Tab. 1b. We could see that our method surpasses the others for all metrics.

## 5 Conclusion and Limitation

In this work, we propose low-rank subspaces to perform controllable generation in GANs. The low-rank decomposition of the Jacobian matrix established between an arbitrary image and the latent space yields a null space, which enables image local editing by simply altering the latent code with no need for spatial masks. We also find that the low-rank subspaces identified from the local

region of one image can be robustly applicable to other images, revealing the internal semantic-aware structure of the GAN latent space. Extensive experiments demonstrate the powerful performance of our algorithm on precise generation control with well-trained GAN models.

As for the limitation, the proposed LowRankGAN is able to precisely control the local region of GAN synthesized images. As discussed in Sec. 4.3, our method is robust to the region of interest. For example, to edit the eyes of faces, we do not need accurate eye segmentation yet only require a rough region around the eyes. This yields a shortcoming of our approach, which is that it almost fails to achieve *pixel-level* control. More concretely, our LowRankGAN can only perform editing by treating eyes as a whole, but fails to edit every individual pixel of eyes. We also find that our algorithm tends to control both eyes simultaneously. As a result, it is hard to close only one eye by keeping the other open.

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
