# Low-Rank Subspaces in GANs
## *Supplementary Material*

**Jiapeng Zhu**[1]  **Ruili Feng**[2,3]  **Yujun Shen**[4]  **Deli Zhao**[2]
**Zheng-Jun Zha**[3]  **Jingren Zhou**[2]  **Qifeng Chen**[1]*
[1]Hong Kong University of Science and Technology  [2]Alibaba Group
[3]University of Science and Technology of China  [4]ByteDance Inc.
{jengzhu0, ruilifengustc, shenyujun0302, zhaodeli}@gmail.com
zhazj@ustc.edu.cn  jingren.zhou@alibaba-inc.com  cqf@ust.hk

This paper proposes the low-rank subspaces to perform controllable image generation in GANs. In this supplementary material, we first give the proof of Theorem 1 in the main paper. Then a detailed introduction about the low-rank factorization and the Alternating Directions Method of Multipliers (ADMM) [8, 1] algorithm in Sec. 2 are presented. Second, the ablation study on the parameter $\lambda$ of Eq. (4) (in the main paper) and $r_{\text{relax}}$ of *precision relaxation* in Sec. 3.3 (in the main paper) is given in Sec. 3. Third, more comparison results with other methods are given in Sec. 4 to show the advantages of our method. Fourth, Sec. 5 gives more results on controlling diverse regions or on other datasets such as MetFace [5]. Fifth, we show that the attribute vectors we find can be easily applied to real images in Sec. 6. At last, we designed an experiment to validate the potential application of our method in Sec. 7.

## 1 Proof of Theorem 1

**Theorem 1.** *Assume that the real data $\mathcal{X}$ lie in some underlying manifold of dimension $dim(\mathcal{X})$ embedded in the pixel space. Let $\boldsymbol{z}_k \in \mathcal{R}^{d_z}$ denote the latent code of the $k$-th layer of the generator of GAN over $\mathcal{X}$ and $\boldsymbol{J}_{z_k}$ be the Jacobian of the generator with respect to $\boldsymbol{z}_k$. Then we have $rank(\boldsymbol{J}_{z_{k+1}}^T \boldsymbol{J}_{z_{k+1}}) \leq rank(\boldsymbol{J}_{z_k}^T \boldsymbol{J}_{z_k}) \leq d_z$, where $rank(\boldsymbol{J}_z^T \boldsymbol{J}_z)$ denotes the rank of the Jacobian. Further, if $dim(\mathcal{X}) < d_z$, then $rank(\boldsymbol{J}_{z_k}^T \boldsymbol{J}_{z_k}) < d_z$.*

*Proof.* Let $Jf$ denote the Jacobian of function $f$. By the chain rule of differential [9], we have

$$rank(J(f^1 \circ f^2)) = rank(Jf^1 Jf^2). \tag{1}$$

Recall that for any two matrices $A$ and $B$,

$$rank(AB) \leq \min\{rank(A), rank(B)\}. \tag{2}$$

Combining them together, we have

$$rank(J(f^1 \circ f^2)) \leq \min\{rank(Jf^1), rank(Jf^2)\}. \tag{3}$$

As $rank(A^T A) = rank(A)$ and the generator function is a coupling of each layer, we then have

$$rank(\boldsymbol{J}_{z_{k+1}}^T \boldsymbol{J}_{z_{k+1}}) \leq \min\{rank(\boldsymbol{J}_{z_k})\}. \tag{4}$$

Theoretically, the converged generator will output the whole data manifold, i.e. $\boldsymbol{G}(\mathcal{Z}) = \mathcal{X}$. Thus, we have $rank(\boldsymbol{J}_{z_0}) = dim(\mathcal{X})$. If $dim(\mathcal{X}) < d_z$, we have $\boldsymbol{J}_{z_k}^T \boldsymbol{J}_{z_k} \leq \boldsymbol{J}_{z_0}^T \boldsymbol{J}_{z_0} < d_z$. □

---

* denotes the corresponding author.

35th Conference on Neural Information Processing Systems (NeurIPS 2021).

## 2 Low-Rank Factorization

Low-rank factorization of a matrix solves the decomposition of a low-rank matrix corrupted with random noise, which has been well studied in the past two decades and widely applied to solve real tasks in various disciplines [11]. Given matrix $M$, its low-rank factorization can be written as

$$M = L_0 + S_0, \tag{5}$$

where $L_0$ is the recovered low-rank matrix and $S_0$ is a sparse noise matrix. Classical PCA seeks the best rank-$r$ estimate of $L_0$ [11] via Singular Value Decomposition (SVD). Namely, if $M = U\Sigma V^*$ is SVD of the matrix $M$, then the optimal rank-$r$ approximation to $M$ is

$$\hat{L} = U\Sigma_r V^T = \sum\nolimits_{i=1}^{r} \sigma_i u_i v_i^T, \tag{6}$$

where $\Sigma_r$ just keeps the first $r$ singular values of the diagonal matrix $\sigma$ and $T$ denotes the transpose. This solution only works well when the noise $S_0$ is small such as Gaussian random noise with small standard deviation. However, neither the scale nor the distribution of $S_0$ is known in advance. And the new problem can be categorized to *Robust PCA*, which is formulated as follows:

**Problem 1.** *Given $M = L + S$, both $L$ and $S$ are unknown, but $L$ is known to be of low rank and $S$ to be sparse. In this scenario, $S$ can have elements with arbitrary magnitude. The task is to correctly recover $L$.*

In particular, we are now seeking $L$ of the lowest rank to reconstruct $M$ with some sparse noise perturbed. More formally, the optimization can be written as:

$$\begin{aligned} \min_{L,S} \quad & \text{rank}(L) + \lambda\|S\|_0, \\ \text{subject to} \quad & L + S = M, \end{aligned} \tag{7}$$

where $\|S\|_0$ is the number of nonzero elements in $S_0$. If this constrained optimization can be solved, both $L_0$ and $S_0$ can be simultaneously obtained accordingly. Unfortunately, Eq. (7) is a highly nonconvex function to be solved. To handle this, the convex relaxation method is exploited to achieve a tractable solution by using the $\ell_1$ norm and the nuclear norm as the surrogates of the sparsity and low-rank [4], respectively. So the problem can be reformulated as follows:

$$\begin{aligned} \min_{L,S} \quad & \|L\|_* + \lambda\|S\|_1 \\ \text{subject to} \quad & L + S = M, \end{aligned} \tag{8}$$

where $\|L\|_* = \sum_i \sigma_i(M)$ is the nuclear norm defined by the sum of all singular values, $\|S\|_1 = \sum_{ij} |M_{ij}|$ is the $\ell_1$ norm defined by the sum of all absolute values in $S$, and $\lambda$ is a positive weight parameter to balance the sparsity and the low rank. Now the problem becomes a convex one, which is also referred to as *Principal Component Pursuit (PCP)* in [3].

There are many approaches to solving this convex optimization problem. The most common one is the *Lagrange multiplier* method by solving an unconstrained minimization problem rather than the original constrained one. For Eq. (8), we can get the augmented Lagrangian as follows:

$$\begin{aligned} \mathcal{L}_\mu(L, S, \Lambda) = \|L\|_* + \lambda\|S\|_1 + \langle \Lambda, L + S - M \rangle \\ + \frac{\mu}{2}\|L + S - M\|_F^2, \end{aligned} \tag{9}$$

where $\Lambda$ is the multiplier of the linear constraint, $\mu > 0$ is the penalty for the violation of the linear constraint, and $\|\cdot\|_F$ is the Frobenius norm. Obviously, the *alternating directions method of multipliers* (ADMM) [8, 1] is applicable to solve Eq. (9), which is given in Algorithm 1.

In Algorithm 1, $S_\tau(x) = sgn(x)max(|x| - \tau, 0)$ is the soft shrinkage operator, and $D_\tau(M)$ denotes the singular value thresholding operator given by $D_\tau(M) = US_\tau(M)V^T$.

---

**Algorithm 1** Alternating Directions Method of Multipliers (ADMM)

---

**Initialize:** $\boldsymbol{S}_0 = 0, \Lambda_0 = 0, \mu = 0$.
**while** not converge **do**
 Compute $\boldsymbol{L}_{k+1} = D_{1/\mu}(\boldsymbol{M} - \boldsymbol{S}_k - \mu^{-1}\Lambda_k)$;
 Compute $\boldsymbol{S}_{k+1} = S_{\lambda/\mu}(\boldsymbol{M} - \boldsymbol{L}_{k+1} - \mu^{-1}\Lambda_k)$;
 Compute $\Lambda_{k+1} = \Lambda_k + \mu(\boldsymbol{L}_{k+1} - \boldsymbol{S}k + 1 - \boldsymbol{M})$;
**end while**
**Output:** $\boldsymbol{L}_* \leftarrow \boldsymbol{L}_k; \boldsymbol{S}_* \leftarrow \boldsymbol{S}_k$.

---

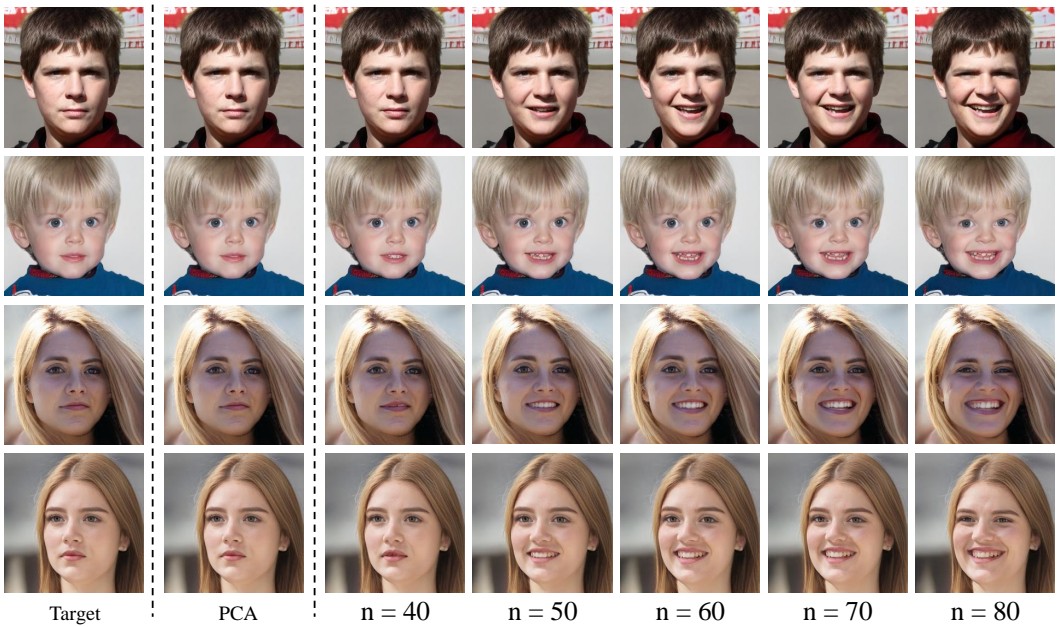

| Target | PCA | n = 40 | n = 50 | n = 60 | n = 70 | n = 80 |

Figure 1: **Ablation study on parameter** $n$. The first two columns represent the target images and the results when projecting the attribute vectors into the "null space" (*i.e.*, eigen vectors regarding small eigen values) of PCA. The rest are the smile editing results with different $n$. Zoom in for details.

## 3 Ablation Study

We perform the ablation study on the parameter $\lambda$ of Eq. (4) and $r_{\text{relax}}$ of *precision relaxation* in this section. First, we give the results on the parameter $\lambda$. Recall that $\lambda$ is a positive weight to balance the *sparsity* and **rank** of the resulting matrices. Hence, a proper value for $\lambda$ is needed since it will influence the null space which we use to project the principal vectors of an edited region. For convenience, we set $\lambda = 1/n$ and study the parameter $n$. Fig. 1 shows the results using different $n$ when adding smile, in which we could find that the larger $n$ we use, the larger degree of editing we could obtain. This is consistent with Sec 4.1 in the main paper since a larger $n$ will result in a larger null space. And a larger null space may contain more components that could affect the region of the mouth in this case. However, a larger $n$ could lead to a change on the other attributes. For instance, the eyes of the man in the first row change a little when $n = 80$. This is because each image has its limit when editing its components. When reaching the limit of the associated attribute, over-editing this attribute will propagate to other regions or attributes. Thus, we choose $n = 60$ since it could achieve the satisfying editing results on the target region and meanwhile could keep the rest region unchanged.

Rigorously, simply conducting PCA on $\boldsymbol{M}$ in Eq.4 in the main paper does not possess a null space since all the eigenvalues are not equal to zeros. To compare with the null space obtained by low-rank decomposition, we choose the eigenvectors corresponding to smallest eigenvalues whose magnitudes are no more than $10^{-7}$) to form an analogous "null space". And then, we project the principal vectors into the null space of PCA. The results are shown in the second column of Fig. 1, in which we could find that after projection, we cannot edit any regions of the target images. This implies that the null space of PCA does not contain any components that could affect the region of the eyes. To be specific, the attribute vectors that could edit the regions of eyes vanishes after projecting them to the null space of PCA. We observe the same phenomenon when editing the other regions. In contrast, the null space by low-rank decomposition contains the components that could mainly affect the regions of eyes yet barely affect the rest region.

Second, we show the influence of the parameter $r_{\text{relax}}$ when editing an image. As shown in Fig. 1 in the main paper, our algorithm aims at manipulating region A with little change in region B. But in some cases, this kind of precise control may not exist, or the degree of editing will be limited due to this constraint. Thus, we involve some attribute vectors with smallest non-zero singular

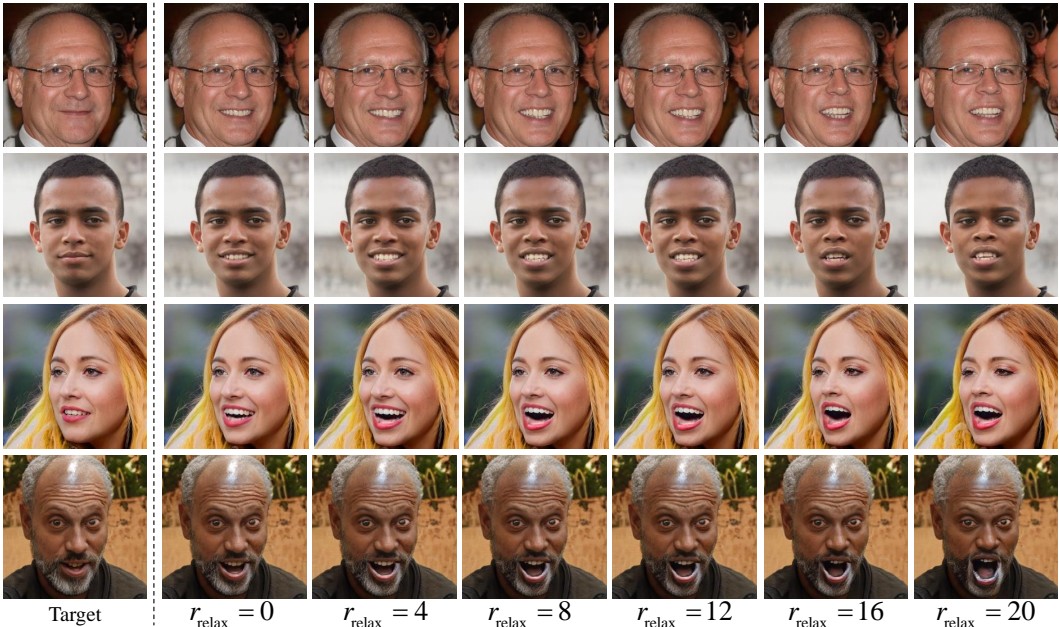

| Target | $r_{\mathrm{relax}}=0$ | $r_{\mathrm{relax}}=4$ | $r_{\mathrm{relax}}=8$ | $r_{\mathrm{relax}}=12$ | $r_{\mathrm{relax}}=16$ | $r_{\mathrm{relax}}=20$ |

Figure 2: **Ablation study on parameter** $r_{\mathrm{relax}}$. The first column shows the target images and the remaining are the smile editing results. Zoom in for details.

values to allow slight change in region B when editing region A. We conduct the experiment on mouth editing to illustrate the influence of $r_{\mathrm{relax}}$. It is obvious that it is impossible to open the mouth naturally without any change on the chin. We can see that in Fig. 2, the mouth opens with a relatively small degree when $r_{\mathrm{relax}} = 0$. And with the increase of $r_{\mathrm{relax}}$, the degree of mouth opening is larger. However, when $r_{\mathrm{relax}}$ becomes larger, the other attributes will change. For example, we could observe a slight change in the identity in the first two rows when $r_{\mathrm{relax}} = 20$, and an unrealistic edited image in the last row. Thus, a number between 4 and 12 could be a good choice since the other attributes could maintain well in this case, and we could also achieve expected editing results.

## 4   Comparison with Existing Methods

Besides the methods we compare in the main text, here we compare our method with the baseline such as feature blending [10] and recently proposed StyleSpace [12]. First, we compare our method with feature blending on closing eyes. We first find a reference image with closed eyes and then replace the corresponding features in the target images using the feature from the reference image. As shown in Fig. 3, simply replacing the features in the related feature maps fails to result in photo-realistic results. The results on $32 \times 32$ are the best ones in all feature blending. However, the reference image and the target images encounter severe misalignment problems in most cases. Instead, the editing results are significantly improved when using the attribute vector obtained from the reference image by our method. For instance, our algorithm successfully closes the eyes of the target images regardless of the pose, gender, etc.

Furthermore, we compare our method with feature blending [10] on closing mouth. We first find a reference image with a closed mouth and then replace the corresponding features in the target images using the feature from the reference image. As shown in Fig. 4, simply replacing the features in the related feature maps also results in unrealistic results. For example, besides the misalignment problem between the reference image and the target images, the results for $32 \times 32$ also suffer from color distortion in the mouth region. On the contrary, the editing results are substantially improved when using the direction obtained from the reference image by our method. For instance, our method successfully closes the mouths of the target images regardless of the pose, gender, age, etc.

We also provide the comparison results with StyleSpace [12] on several attributes such as closing and opening eyes, closing and opening mouth, etc. Fig. 5 shows the comparison results on closing and

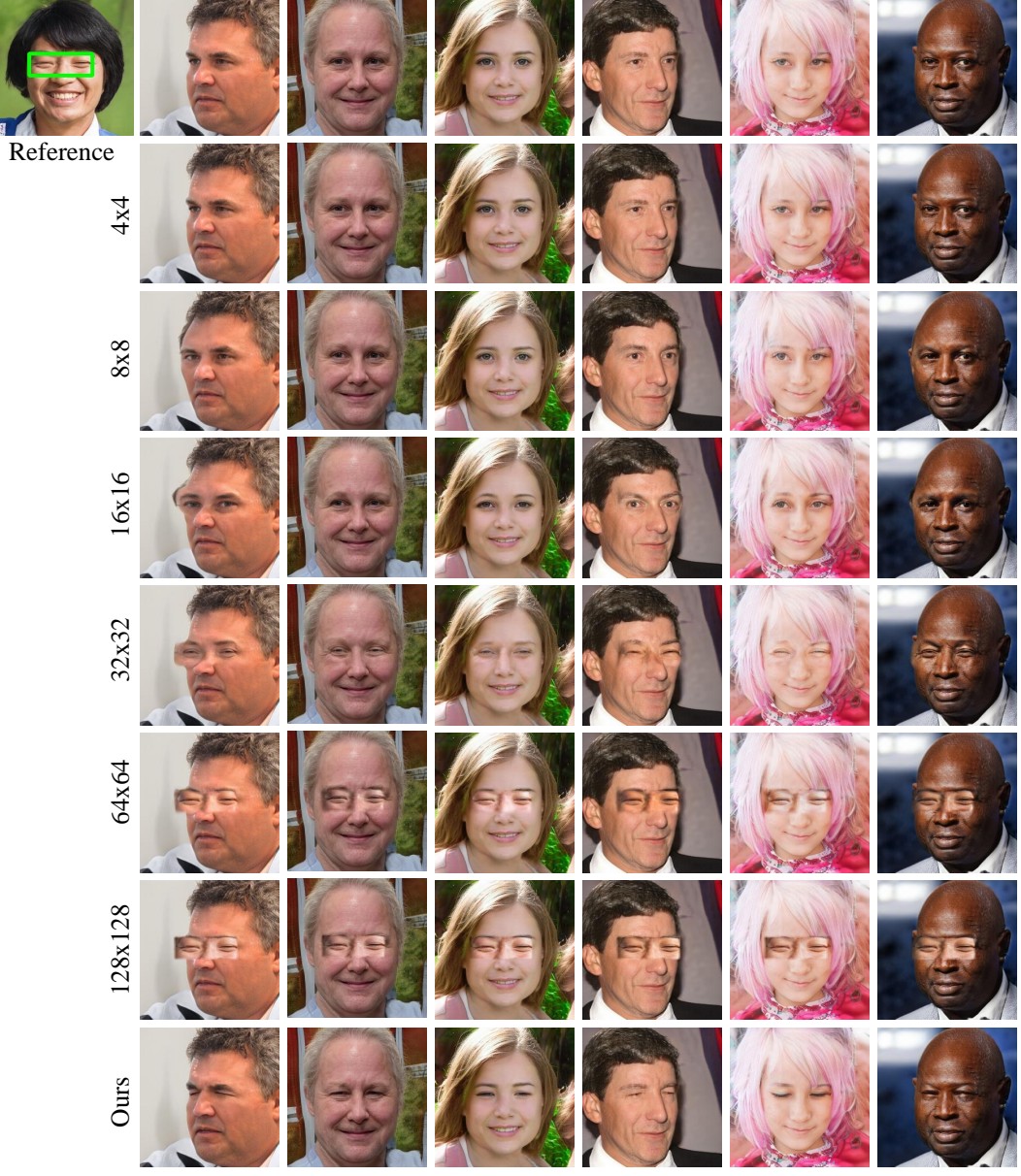

Figure 3: Comparison with feature blending method [10] on closing eyes. The top-left image is the reference image under a particular bonding box. The top row are the target images for editing and the bottom row are the results with our LowRankGAN. The other rows show the results from [10] by blending features at different resolutions. Zoom in for details.

opening eyes, in which we could find that our methods could achieve more photo-realistic results compared with StyleSpace [12]. For instance, the artifacts appear when closing eyes using StyleSpace. The color of the eyeball turns to be white when opening eyes. Fig. 6 shows the comparison results on closing and opening the mouth, in which we could find that both StyleSpace and our method could successfully close or open the mouth of the image. However, there are subtle differences between them. For example, the jaw becomes narrow when closing the mouth and wide when opening the mouth using the method proposed in StyleSpace. This is against common sense since the jaw will not be widened simultaneously when opening the mouth of a person. Actually, such cases are common when manipulating on the feature maps since it may be easily overfitted. Rather, our method could achieve satisfying results consistent with photo-realism and visual perception.

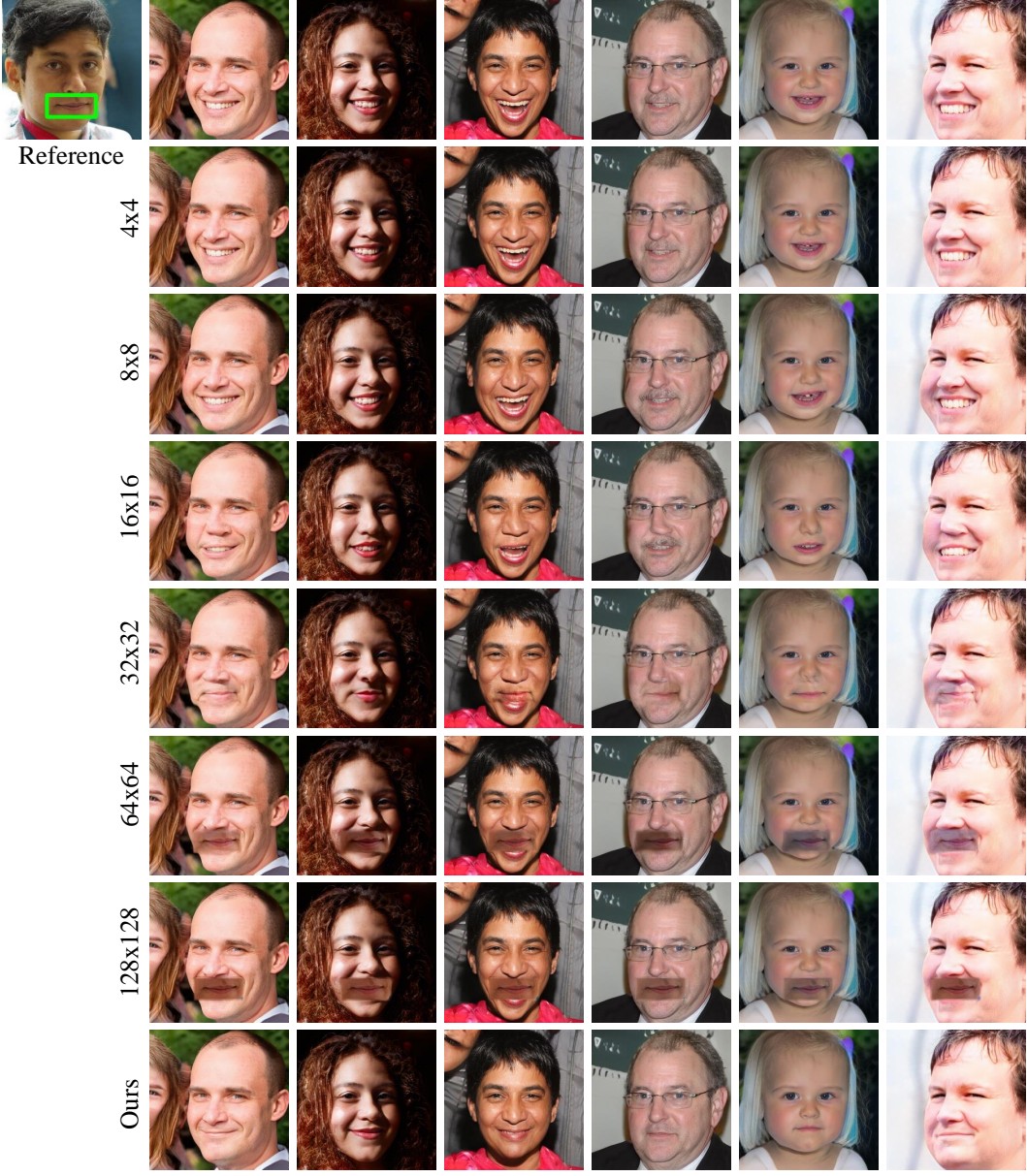

Figure 4: Comparison with feature blending method [10] on closing mouth. The top-left image is the reference image under a particular bonding box. The top row are the target images for editing and the bottom row are the results with our LowRankGAN. The other rows show the results from [10] by blending features at different resolutions. Zoom in for details.

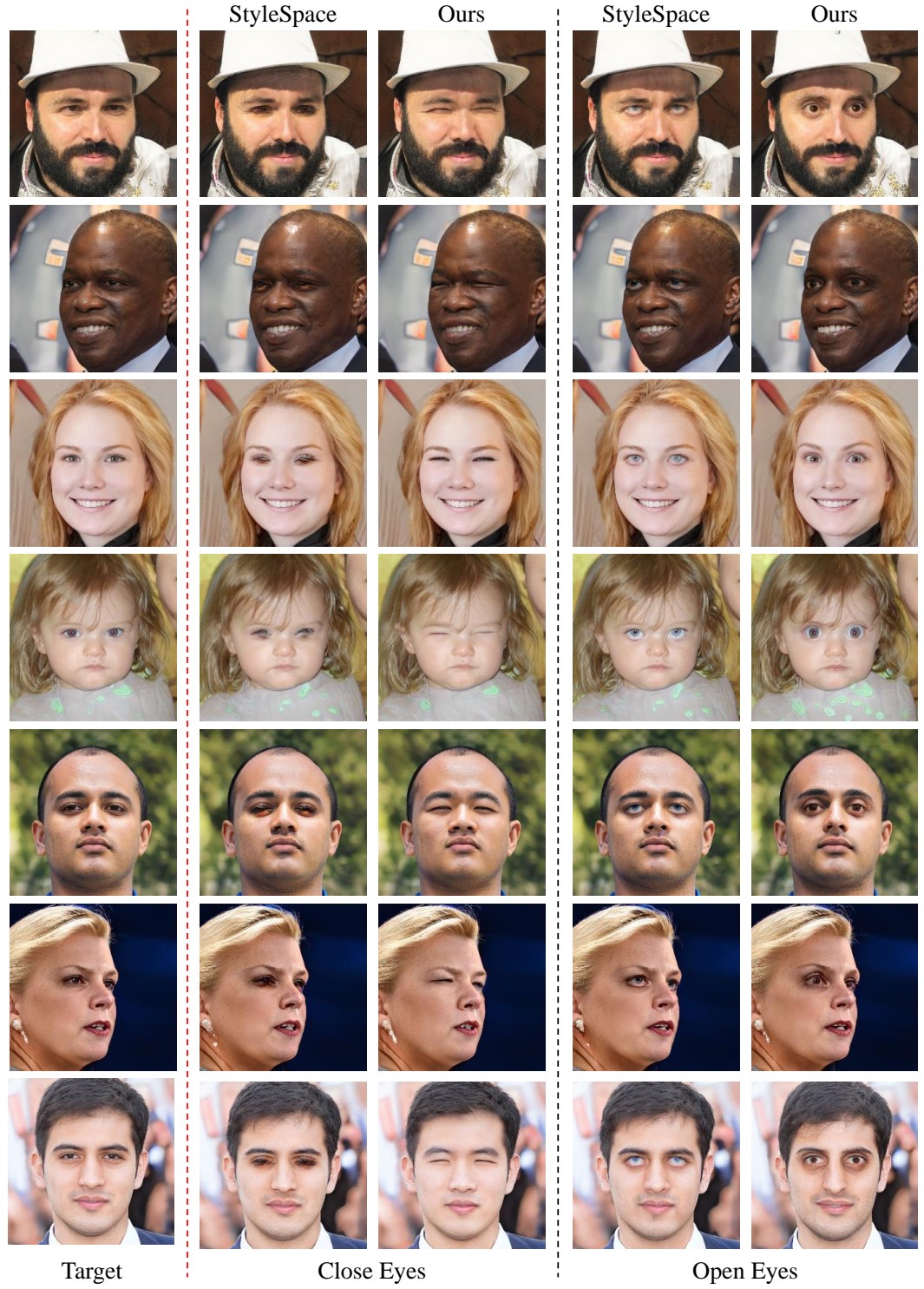

Figure 5: Comparison with StyleSpace [12] on eye attributes. Zoom in for details.

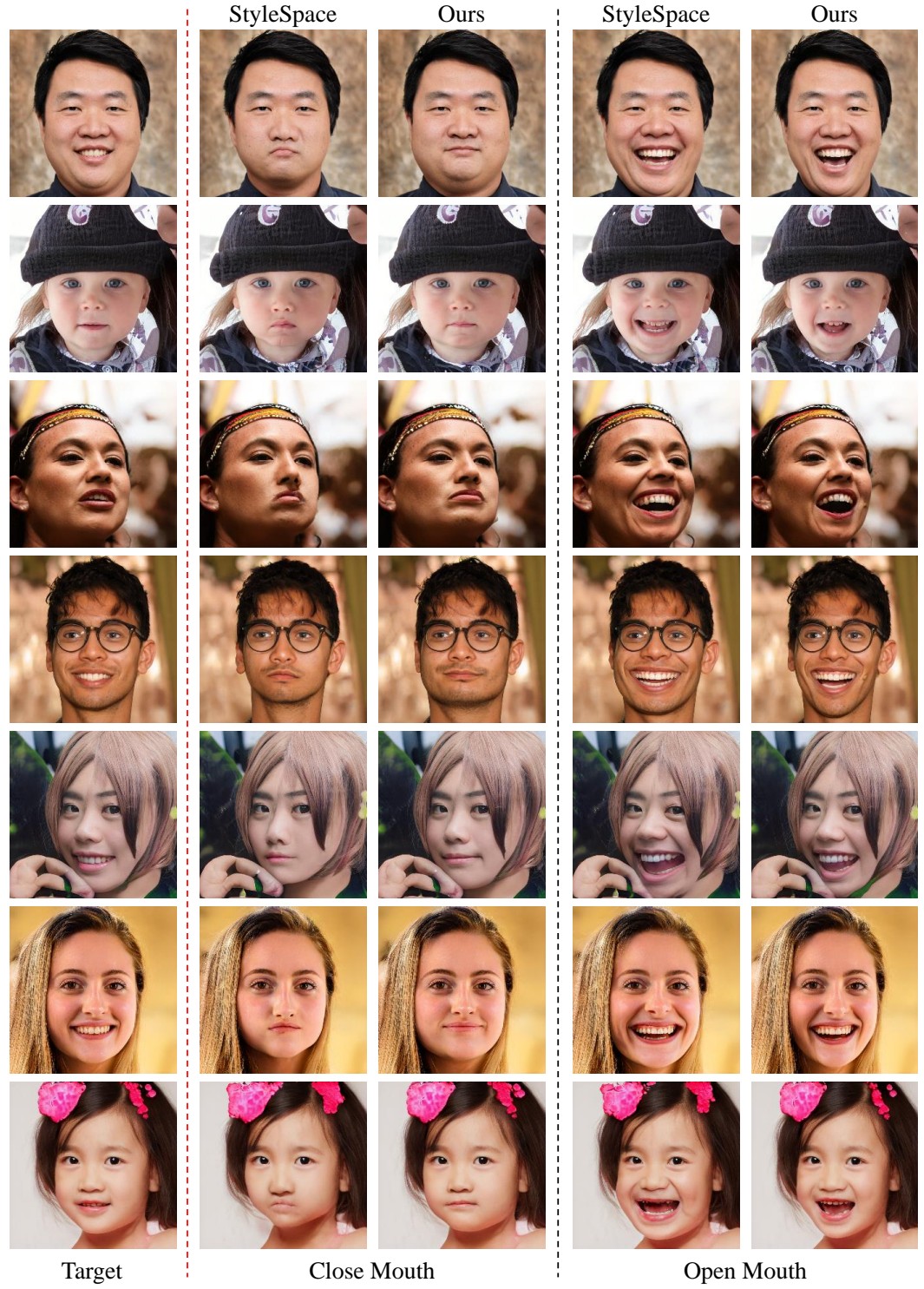

StyleSpace    Ours    StyleSpace    Ours

Target    Close Mouth    Open Mouth

Figure 6: Comparison with StyleSpace [12] on mouth attributes. Zoom in for details.

# 5 More results

This section provides more results about the experiments in the paper such as the comparison on null space projection in Fig. 7 and generalization from one image to others in Fig. 8. As shown in Fig. 7, the remaining region changes violently without projection. For example, the woman's face becomes smaller when closing the mouth, and there appear some artifacts when closing her eyes. For MetFace [5] in the second row, the pose is changed when closing the eyes. And for the butterflies in the third row, the purple flowers disappear when the size gets smaller, the red twig turns purple, and the shape of the leaf also changes. Instead, those details are well preserved when using null space projection. As shown in Fig. 8, we show the results on the other datasets such as on MetFace [5] and LSUN church [13] or the other attribute such as zoom-in via BigGAN, where we could also verify the advantages summarized in Sec. 4.2 of the main text.

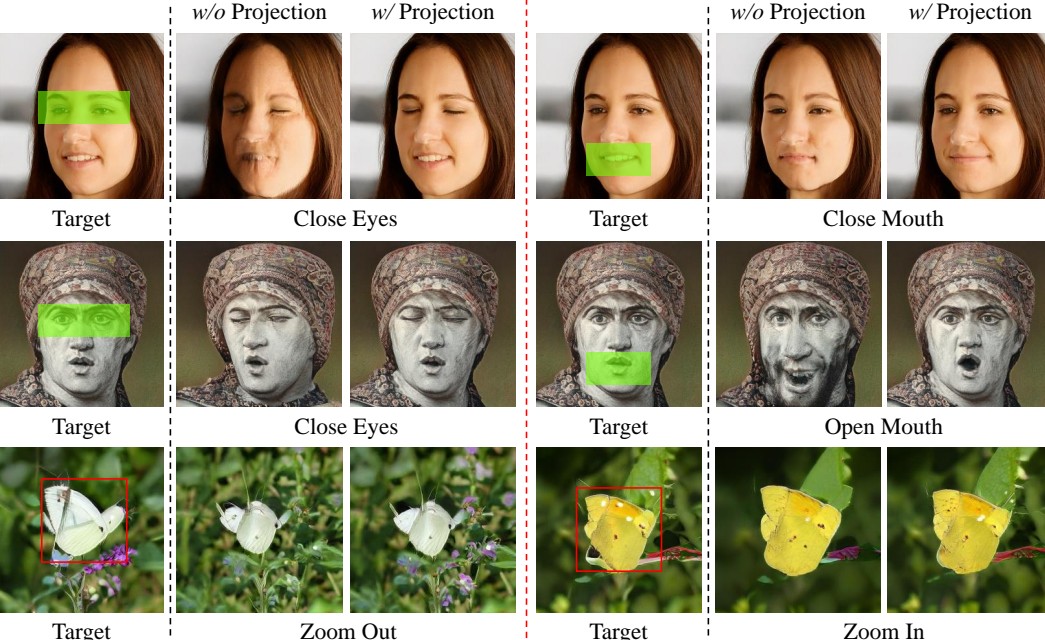

Figure 7: **Precise generation control** via the null space projection on StyleGAN2 [6] and BigGAN [2]. For each image attribute, the original attribute vector identified from the region of interest (under **green** masks or **red** boxes) acts as changing the image globally (*e.g.*, the face shape change in the top row and the background change in the bottom row). By contrast, our approach with null space projection yields more satisfying local editing results.

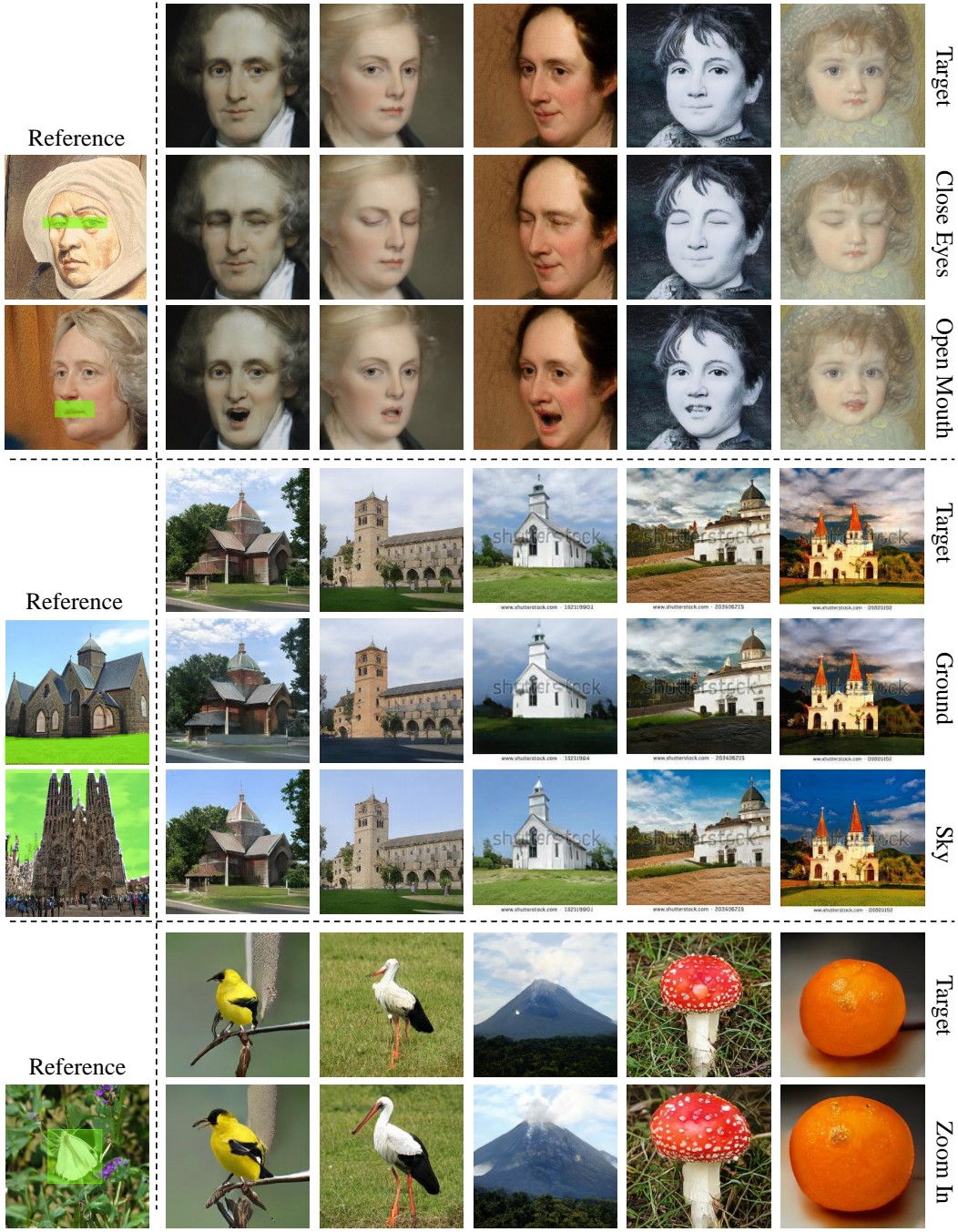

Figure 8: Generalization of the latent semantics identified from the local region of one synthesis to other samples on MetFaces (StyleGAN2), LSUN church (StyleGAN2), and ImageNet (BigGAN). Our approach does not require the target image to have the same masked region as the reference image. For example, the sky and the ground of the church can have different shapes. The results on BigGAN even suggest that the zoom-in semantic identified from one category can be applied to other categories.

# 6 Results on Real Images

Given the attribute subspaces obtained by low-rank decomposition, we can easily apply them to real images. First, we need to project the real images into the latent spaces of the GANs, i.e. the GAN inversion. Several works [6, 14] have been developed to deal with this problem. Here we employ the method in [6] to project the real images into the $w$ space in StyleGAN2. From Fig. 9, we can see that our method successfully edits the attributes on real images such as opening the eyes or closing the eyes, opening the mouth or closing the mouth, which demonstrates the feasibility of our method to real image editing.

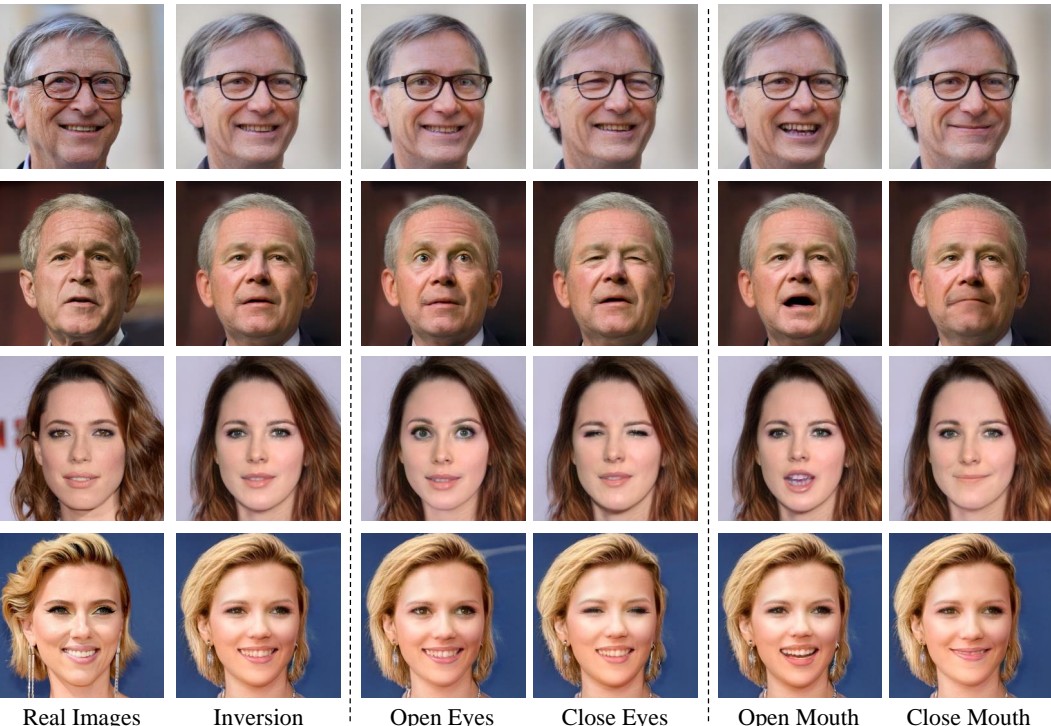

| Real Images | Inversion | Open Eyes | Close Eyes | Open Mouth | Close Mouth |

Figure 9: **Real image editing**. From left to right: real images, inversion results, and the editing results with LowRankGAN regarding different attributes.

# 7 LowRankGAN for Data Augmentation

As stated in Sec. 4.2 of the main paper, a potential application of our algorithm could be data augmentation. Hence,in this section, we conduct an experiment to further support this claim.

Our experiment aims at training a classifier to determine whether the person in an image is closing eyes or not, and the original dataset we use is FFHQ. There are $70,000$ face images in FFHQ, among which we find $564$ images with closed eyes. Note that we use a face parsing model [7] to assign the "ground-truth" label to the entire set. Then, we single out $100$ images with open eyes and $100$ with closed eyes, forming a validation set, and treat the remaining $69,800$ samples as the training set. We train a bi-classification model (with ResNet-18 backbone) on *such an imbalanced training dataset*. The converged model gives $52\%$ accuracy on the validation set.

To balance the training set, we use LowRankGAN to synthesize $29,800$ closed-eyes samples (e.g., randomly sampling latent codes and moving the latent codes towards the closing-eye boundary) and replace $29,800$ samples from the original training set. Note that the GAN model is also trained on FFHQ, meaning that we do not use additional closed-eyes data in the entire training process (even when training the GAN model). Hence, the comparison is fair. After the replacement, we train the bi-classification model with the same structure and training hyper-parameters (e.g., batch size and learning rate). The new model improves the accuracy from $52\%$ to $95\%$.

From the above experiments, we can see that LowRankGAN indeed has the potential to augment the real data, especially for rare samples. This will be of great importance to alleviate some long-tail issues in the industry.