# OpenReview forum: "Low-Rank Subspaces in GANs"
_NeurIPS.cc/2021/Conference — NeurIPS 2021 Poster_

### Official Review · Reviewer_HHnz · 2021-07-10

**Rating:** 7
**Confidence:** 3

**Summary:**

This paper proposes using low-rank factorization of the Jacobian of generator’s selected output region with respect to its latent noise source in order to perform precise edits of generated images. Specifically, a latent vector in the top-subspace (imply maximium edit impact) of the output region is identified, after which, it is then projected onto the null-space (no edit impact) of the negative region (the region not selected).

Contributions:
1) Examine the low-rank subspaces of of the generator output w.r.t. the latent space and show how to perform edits in an unsupervised manner.
2) Demonstrate that this process need not be repeated on each image individually, but instead can be performed just once and generalize to many images due to the semantic organization of the generator’s latent space.
3) Perform several experiments on various image datasets, comparing against other state of the art models, both qualitatively and quantitatively.

**Limitations And Societal Impact:**

I included a suggestion for discussion of a specific limitation in my review above. The authors might also want to add a comment on the use of GAN editing for producing fake news among other things.

**Main Review:**

Originality:
To my knowledge, this approach is novel and also appears to be much simpler than previous approaches.

Quality: The paper is of good quality.

Clarity: There are some small typos in places, but otherwise the paper is clearly written.

Significance: Precise editing of generated / synthetic images is of great important to the computer vision and graphics communities.


Overall, I liked the simplicity of the approach and the results in several cases are impressive. To improve the paper, I would suggest:
- Figure 2 is visually a bit disappointing relative to the rest of in the paper. What are the units on the $l_1$-loss heatmaps? Is it the sum of the absolute values of the differences across the rgb channels $[0, 255]$? I can see the cloud is slightly larger in (b) than (a), but I can’t visually tell the any difference between (d) and (e). Please comment on any visual difference we should be noticing.
- Figure 3 (bubble): In order to confirm that the “white dots” were not there before, we need to see a zoomed-out version of the target image.
- Could you include Figure 4 repeated but with $v$ (the original attribute vector) instead of $p$ (the vector projected into the null-space)?
- Figure 6 is interesting but reveals a downside to the approach. The fact that different masks lead to similar edits means a lack of precise control of edits. Can you please discuss this limitation?


Minor:
- I don’t have a counterexample, but this sentence raised alarm bells for me: “first to demonstrate effectiveness of low-rank factorization in deep generative models”. It seems like an overstatement.

**Time Spent Reviewing:**

1.5

---

> ### Author Response · Authors · 2021-08-10
> **Response to Reviewer HHnz**
>
> Thanks for the valuable comments. Individual concerns are addressed as follows.
>
> **Q1. Explanation on the large difference between (d) and (e) in Fig. 2.**
>
> Yes, the heatmap is computed by the $\ell_1$ differences across the RGB channels, and all pixel values are in range $[0, 255]$. The large difference is caused by *a slight pixel shift*, which may be hard to recognize by simply comparing two images. Hence, we attach a video from **[this anonymous link](https://ufile.io/2q9fgtr4)** for better visualization. Thanks for pointing this out. We will use a better example and a large manipulation strength in the next version, but the conclusion that *vector within the null space barely affects the region of interest* still holds.
>
> **Q2. About zooming in the bubble in Fig. 3.**
>
> LowRankGAN focuses on local editing, which is achieved by dividing an image into the region of interest (A) and the remaining region (B), and projecting A-related vectors onto B-derived null space. In this way, only A region will be edited and B region should be kept untouched. Consequently, the zooming-in operation should only focus on the foreground (*i.e.*, the bubble) while the background should be preserved.
>
> **Q3. About the comparison between the original attribute vector and the vector projected into the null space in Fig. 4.**
>
> Please refer to **[this anonymous link (Fig. 6, 7)](https://ufile.io/mi1jbf7b)** for the comparison between before and after null space projection.
>
> **Q4. About Fig. 6 in the paper: lack precision control, discuss it.**
>
> Please refer to **Limitation and Social Impact** in common concerns.

---

> > ### Comment · Reviewer_HHnz · 2021-08-31
> > **Thank you for your explanations**
> >
> > I maintain my score.
> >
> > Also, after reading your response to xDMm's review, I agree that [24] is the closest related work in terms of leveraging the Jacobian of the generator w.r.t. $z$ to traverse the latent space. That being said, the right singular vectors of $J$ **are** the eigenvectors of $J^\top J$, i.e., identifying the null-space of $J^\top J$ is the same as identifying the right singular vectors of $J$ that have zero singular value. You may want to make this connection more clear. But I agree, [24] focuses on the "top-k" singular vectors, while you focus on the "bottom-k".

---

> > > ### Author Response · Authors · 2021-09-01
> > > **Thanks for the comments**
> > >
> > > Thanks for liking our work and also thanks for your effort in reviewing our paper and participating in the discussion. You are right that computing the right singular vectors of $J$ is the same as computing the eigenvectors of $J^TJ$. But, there is no guarantee that the null-space of $J^TJ$ exists. To make sure finding the semantics that only affect region A instead of region B, we propose to project eigenvectors of $J_A^TJ_A$ onto the null-space of $J_B^TJ_B$. This is the key step of LowRankGAN. From this perspective, the low-rank factorization is essential to ensure the null-space of $J_B^TJ_B$. We will make this clear and thanks for your suggestion.
> > >
> > > Meanwhile, besides the low-rank factorization and projection, another major difference between our work and [24] is that we demonstrate "the semantics found from *only one* image, by computing Jacobian only once, can be convincingly applied to others with no effort." Instead, [24] requires computing the Jacobian at each step. We will also add the discussion.

---

### Official Review · Reviewer_xDMm · 2021-07-12

**Rating:** 4
**Confidence:** 4

**Summary:**

The authors propose an approach to identify meaningful latent directions in a GAN latent space. First, they factorize the Jacobian of a generator map $G: z \to \mathrm{RGB}$, treating its principal axis as the desired directions. Then they fine-tune them in a supervised regime to guarantee only the regions of the interest being edited. Though some of the results look intriguing, a lot of relevant comparisons with prior works is missed and the experiments are not sufficient.

**Limitations And Societal Impact:**

There is no limitations section.

**Main Review:**

Strengths:
- some of the results look promising, especially, the transferability of the directions discovered by a single image to the whole dataset.

Weakness:
- As for me, it was really hard to read the paper. Its structure is unclear, the introduction gives just a vague idea of the contribution. The experiments have no implementation details which makes them unreproducible as the authors do not supply the code.
- Though the authors state that one of the main contributions is the locality of the editings produced by their directions, the comparison with the prior works is unfair as they use additional supervision provided by either segmentation masks or region boundaries. It would be reasonable to provide the editing comparison of the proposed approach with e.g. [1, 2].
- It remains unclear how a direction discovered for a particular $z$ is translated to the whole latent space. Do the authors use always the same z to discover the interpretable directions?
- On line 85 the authors state that "to our best knowledge, we are the first to demonstrate the effectiveness of low-rank factorization...". The very close idea to utilize a generator's Hessian eigenvectors as interpretable directions was first proposed in [3] which is not cited. A close idea to use SVD or Hessian-based editable weights decomposition was also discussed in [4].
- line 157: it is unclear, how exactly the authors determine if a particular vector influences region $A$.
- The only quantitive evaluation is presented in Table 1. The authors perform a comparison with unsupervised approaches ignoring supervision approaches. It is also unclear, how do they scale the magnitudes of the latent shifts for different methods. For a particular latent direction $h$, I would be happy to see a plot of points [ $\mathrm{FID}(G(z + \alpha \cdot h))$, $\mathbb{E}~ \mathrm{LPIPS}(G(z), G(z + \alpha \cdot h))$ ] with different shift scales $\alpha$. The first coordinate is the FID of the shifted latent distribution and the second coordinate is the expectation of an image deformation in terms of the lpips-distance.
- line 251: "Thereby, a potential application of our algorithm could be data augmentation." it would be preferable if the authors could demonstrate this experimentally.

[1] Editing in Style: Uncovering the Local Semantics of GANs, Edo Collins et al.

[2] Mask-Guided Discovery of Semantic Manifolds in Generative Models, Mengyu Yang et al.

[3] The Hessian Penalty: A Weak Prior for Unsupervised Disentanglement, William Peebles et al.

[4] Navigating the GAN Parameter Space for Semantic Image Editing, Anton Cherepkov et al.

---------
UPD: I raise my score 3 -> 4
I still think that the quantitive evaluation is insufficient as it consists of a table in the main text with only two attributes and one figure in the response for a single attribute. In my opinion, some of the baselines (though, with higher supervision) should also be reported.

**Time Spent Reviewing:**

6

---

> ### Author Response · Authors · 2021-08-10
> **Response to Reviewer xDMm**
>
> Thanks for the valuable comments. Individual concerns are addressed as follows.
>
> **Q1. About the implementation details and the code.**
>
> Our algorithm is easy to implement, which is appreciated by the other two reviewers. Concretely, our algorithm mainly consists of three steps, which are computing Jacobian (Sec. 3.1), low-rank factorization (Sec. 3.2), and null space projection (Sec 3.3). All these steps have the corresponding mathematical formulation. We have two hyper-parameters in total, which are $\lambda$ and $r_{relax}$. The ablation studies on these two hyper-parameters can be found in the supplementary material. The manipulation model (Eq. (1)) is commonly used in prior work [14, 17, 25, 26]. We will discuss the manipulation strength in the next version. Also, the code will be released once the paper gets accepted.
>
> **Q2. Lacking the comparison with other methods.**
>
> LowRankGAN only uses a *rough bounding box* for local editing. We argue that the rough bounding box cannot be viewed as a supervision, because to achieve local editing, users must provide a region of interest to edit, or otherwise, the "local" is hard to define. Unlike "Editing in Style" [8] which requires a reference image for local editing (for example, if a user wants to use [8] for closing human eyes, he/she should first find an image with closed eyes), our LowRankGAN discovers the manipulation direction automatically. As a result, LowRankGAN is done in a completely unsupervised manner.
>
> Meanwhile, beyond comparing with GANSpace [14] and SeFa [25], we also **have compared with feature blending approaches [9 (supp), 11 (supp)] on image local editing** in the supplementary material (Fig. 3, 4, 5, 6). Note that [11 (supp)] is a recent paper published in CVPR'21, which achieves the state-of-the-art performance. We will move some results into the main paper.
>
> **Q3. It remains unclear how a direction discovered for a particular $z$ is translated to the whole latent space. Does the author always use the same $z$ to discover the interpretable direction?**
>
> Yes, for the experiments on each semantic (like hairs or eyes), we only use one $z$ to find the directions. Also, in all figures, the reference images used to discover the interpretable directions are visualized together with the target images.
>
> **Q4. Claim on "the first to demonstrate the effectiveness of low-rank factorization...". Hessian, SVD have been used in previous work.**
>
> **Disagree. This must be a misunderstanding.** Low-rank factorization has significant differences from SVD and Hessian. The **null space derived by the low-rank factorization** is the core on why our approach can achieve **local editing**. We give detailed explanation as follows:
>
> First, we clarify the misunderstanding between Jacobian and Hessian. The former is the first-order derivative, while the latter is the quadratic derivative. Besides, replacing Jacobian with Hessian in Eq. (3) of our paper is incorrect. This equation is the basis for our paper.
>
> Second, the purpose of using these two matrices is also different. The Hessian penalty mainly focuses on the disentanglement when training GANs (*i.e.*, the output of the generator becomes smoother and more disentangled in the latent space). Instead, our work focuses on **precise control of a specific region of the generated images on pre-trained GANs**.
>
> Third, the low-rank factorization is essentially different from SVD. The purpose of the low-rank factorization is to find the intrinsic low-rank representation of a matrix. Rather, SVD is used to decompose a matrix. The relationship of those two operations is listed in the paper (Eq. (4) and Eq. (6)), which are the key and main content of our paper.
>
> At last, the first work of using SVD on Jacobian is [24], which we have already cited and discussed in our paper. But contrary to their paper, which computes the right-singular vectors of $J$ on *each* editing step, we perform low-rank factorization on $J^TJ$, and just compute the Jacobian once. Moreover, we propose **local editing with null space projection**, which is not explored by existing work.
>
> We will add other missing references.
>
> **Q5: It is unclear how exactly the authors determine if a particular vector influences region A.**
>
> This is clearly explained in Sec. 3.1. Based on Eq. (3), the eigenvectors corresponding to larger eigenvalues have a larger influence on the output, and eigenvectors with zero eigenvalues barely influence the output. This is naturally implied by Jacobian.
>
> **Q6. Quantitative results with respect to the manipulation strength.**
>
> Good suggestion. The requested curve can be found in **[this anonymous link (Fig. 5)](https://ufile.io/mi1jbf7b)**. We can tell that our LowRankGAN surpasses GANSpace [14] and SeFa [25] on both precise image control (masked mean square error) and image quality (FID).
>
> **Q7. Potential application on data augmentation.**
>
> Please refer to **LowRankGAN for Data Augmentation** in common concerns.

---

> > ### Comment · Reviewer_xDMm · 2021-08-25
> > **discussion**
> >
> > Thanks a lot for the detailed response.
> >
> > I have a couple of questions and would be pleased if the authors could provide some details:
> >
> > - I would appreciate if the authors could point me the way $r_a$ and $r_b$ (lines 157-159) are chosen for a particular task?
> >
> > - How the reference $z$ is chosen? Is it just an arbitrary random latent?

---

> > > ### Author Response · Authors · 2021-08-26
> > > **Response**
> > >
> > > Thanks for reading the response. The answers to your questions are listed as follows:
> > >
> > > **Q1. Describe the way $r_a$ and $r_b$ (lines 157-159) are chosen for a particulate tasks.**
> > >
> > > Thanks for pointing this out. The $r_a$ and $r_b$ are determined by the hyper-parameter $\lambda$ in Eq. (4). Concretely, given a matrix $M$ to perform low-rank factorization, as well as the hyper-parameter $\lambda$ (which we use $\lambda = \frac{1}{60}$ in our work, see Sec. 3 of the supplementary material), the factorization can be solved by Principal Component Pursuit (PCP), resulting in $L$ and $S$. Consequently, $r_a$ and $r_b$ are the rank (*i.e.*, the number of non-zero eigenvalues) of $L$ matrix correspond to region A and region B respectively. From this point of view, $r_a$ and $r_b$ are completely determined by the Jacobian-guided matrix $M = J^TJ$ (which is determined by the selected latent code and the region of interest) and $\lambda$ (which is $\frac{1}{60}$). Intuitively, larger $\lambda$ leads to larger $r_a$ and $r_b$. We will make this clear.
> > >
> > > Also, the larger $r_a$ is, the higher dimension the low-rank subspace will have. That means we can find more steerable directions related to the region of interest. Meanwhile, the larger $r_b$ is, the lower dimension the null space ($d_z - r_b$) will have. Therefore, the projected attribute directions will have fewer diversity (*e.g.*, the region of interest will have fewer variations). The ablation study can be found in Sec. 3 of the supplementary material.
> > >
> > > **Q2. How the reference z is chosen? Is it just an arbitrary random latent?**
> > >
> > > Yes, you are right. Both the reference $z$ (which is used to compute the Jacobian and perform low-rank factorization), and the target image (which is used to evaluate the manipulation performance) are randomly sampled. We have also conducted experiments by using different reference latent codes, yet with the same region of interest, to find steerable dimensions. It turns out that our algorithm is **strongly robust** to the *randomly* selected reference code in that they tend to give similar results. The experimental results in Fig. 4 also suggest that the semantics found by our Low-Rank GAN from one image can be convincingly applied to arbitrarily sampled target latent codes. We will discuss more.

---

### Official Review · Reviewer_6ack · 2021-07-12

**Rating:** 7
**Confidence:** 4

**Summary:**

The authors propose a method to exploit the low-rank structure of the Jacobian of a deep generative model to perform precise image editing. The Jacobian corresponding to a region of an image is constructed and its null space is computed. By varying the input to the generator in the null space along the eigenvectors, only one attribute can be changed at a time, leaving the rest of the image unchanged. Experiments show that the method outperforms strong baselines in terms of semantic editing.

**Limitations And Societal Impact:**

I am not that the authors have discussed limitations of their work. Perhaps they can outline it in the rebuttal and add them to the paper.

**Main Review:**

I think this paper has many strengths:
- The overall method is quite simple and easy to implement.
- The method is unsupervised which is attractive.
- That finding the relevant directions in the latent space using only image, and using it for other images is very interesting and could be useful.
- Both the qualitative and quantitative results are very promising. The experiments have been well done in general, the user study in particular is an important addition to understand image quality. Editing is shown on many different attributes on several datasets.
- The paper is well-written and easy to understand.

Questions and suggestions for improvement:
- I understand the algorithm and why the eigenvectors are calculated. However, what is the guarantee that every eigenvector corresponds to a single semantic attribute? I can easily imagine a case where the eigenvector could change both the hair color and the eye color, for example, depending on the correlations in the dataset. In other words, what are the assumptions made on the dataset used to train the generator?

- What is the step size taken in the direction of the eigenvectors? Can the authors show results where the step size goes from a larger negative number to a larger positive number in an experiment involving zooming, pose changes and eye opening/closing?

- The experimental results in Figure 5 are very interesting and the author mention that this suggests that the method could be used for data augmentation. Would it be possible for the authors to substantiate this claim with a small experiment. This can make the paper even stronger.

**Time Spent Reviewing:**

5 hours

---

> ### Author Response · Authors · 2021-08-10
> **Response to Reviewer 6ack**
>
> Thanks for the valuable comments. Individual concerns are addressed as follows.
>
> **Q1. What is the guarantee that every eigenvector corresponds to a single semantic attribute?**
>
> Like most existing unsupervised GAN semantic discovery approaches, *e.g.*, GANSpace [14] and SeFa [25], the exact meaning of each discovered semantic should be *post*-annotated. Also, similar to [14] and [25], all the eigenvectors found by LowRankGAN are orthogonal to each other. More importantly, different from [14] and [25], the low-rank factorization introduces **null space** (corresponding to zero eigenvalues) such that moving latent codes within such null space will not affect the region of interest. This is guaranteed by the zero eigenvalue. Consequently, if we project the hair-related vector onto the eye-related null space, the projected vector will not affect the eye region. We will emphasize this after Eq. (7).
>
> **Q2. The results on manipulation strength (alpha change from positive to negative).**
>
> Thanks. It is indeed helpful for readers to better understand the manipulation process by including results with gradually varying strength. Generally, we control the strength within the range $[-3, 3]$ for StyleGAN2, and within the range $[-1, 1]$ for BigGAN. Please see **[this anonymous link (Fig. 1, 2, 3, 4)](https://ufile.io/mi1jbf7b)** for the suggested qualitative results. We will add them to the revised version.
>
> **Q3. Substantiate this claim on data augmentation with a small experiment.**
>
> Thanks. We have conducted an experiment on using LowRankGAN for data augmentation as suggested. Please refer to **LowRankGAN for Data Augmentation** in common concerns.

---

> > ### Comment · Reviewer_6ack · 2021-08-30
> > **Thank you for the response, I vote for acceptance**
> >
> > Thanks to the authors for addressing all my questions and providing substantial results in the rebuttal which provides enough evidence that the proposed method is a good one for semantic image editing. The authors should add these results at least in the supplement if the paper is accepted.
> >
> > I have also read the other reviews and the corresponding author's responses.
> >
> > I think the authors have clearly addressed all of Reviewer xDMm's concerns and I hope xDMm can increase their rating to a positive one. I think the paper has enough novelty and promise to be accepted into the conference.

---

> > > ### Author Response · Authors · 2021-09-01
> > > **Thanks for your effort**
> > >
> > > Thanks for liking our work and also thanks for your effort in reviewing our paper and participating in the discussion. We will carefully prepare the revision by properly adding the discussion and experimental results in the rebuttal.

---

### Author Response · Authors · 2021-08-10
**Common Concerns (Rebuttal)**

We thank all the reviewers for their time, effort, and valuable comments. Here, we first summarize the reviews as follows.

Overall, we are encouraged that both **Reviewer 6ack** and **Reviewer HHnz** find our work (which utilizes Low-Rank factorization for unsupervised semantic discovery in pre-trained GAN models) novel, simple and effective, easy to follow, promising and impressive, and of great importance to the community. We also thank all the reviewers for finding the problems in the submission, like some missing details (*e.g.*, manipulation step size), lack of discussion on limitation and social impact, some to-be-improved visualization figures, and how LowRankGAN can be used for data augmentation. Before answering the questions from each reviewer, we would like to first address some common concerns.

### LowRankGAN for Data Augmentation

We are glad that the reviewers are interested in our experiments shown in Fig. 5 of the submission as well as the potential application of LowRankGAN in data augmentation. Following the suggestions from the reviewers, we conduct an experiment to further support this claim.

Our experiment aims at training a classifier to determine whether the person in an image is closing eyes or not, and the original dataset we use is FFHQ. There are $70,000$ face images in FFHQ, among which we find $564$ images with closed eyes. Note that we use a face parsing model to assign the "ground-truth" label to the entire set. Then, we single out $100$ images with open eyes and $100$ with closed eyes, forming a validation set, and treat the remaining $69,800$ samples as the training set. We train a bi-classification model (with ResNet-18 backbone) on *such an imbalanced training dataset*. The converged model gives $52$\% accuracy on the validation set.

To balance the training set, we use LowRankGAN to synthesize $29,800$ closed-eyes samples (*e.g.*, randomly sampling latent codes and moving the latent codes towards the closing-eye boundary) and *replace* $29,800$ samples from the original training set. Note that the GAN model is also trained on FFHQ, meaning that we do not use additional closed-eyes data in the entire training process (even when training the GAN model). Hence, the comparison is fair. After the replacement, we train the bi-classification model with the same structure and training hyper-parameters (*e.g.*, batch size and learning rate). **The new model improves the accuracy from $52$\% to $95$\%.**

From the above experiments, we can see that LowRankGAN indeed has the potential to augment the real data, especially for rare samples. **This will be of great importance to alleviate some long-tail issues in the industry.** We will add this experiment into the revised version and make some discussions.

### Limitation and Social Impact

The proposed LowRankGAN is able to precisely control the local region of GAN synthesized images. As discussed in Sec. 4.3 of the submission, our method is robust to the region of interest. For example, to edit the eyes of faces, we do not need the accurate eye segmentation yet only requires a rough region around the eyes. This yields a shortcoming of our approach, which is that it almost fails to achieve *pixel-level* control. More concretely, our LowRankGAN can only perform editing by treating eyes as a whole, but fails to edit every individual pixel of eyes. We also find that our algorithm tends to control both eyes simultaneously. As a result, it is hard to close only one eye by keeping the other open.

As for the social impact, our work provides a promising tool that can help people with photo editing (especially local editing). However, every coin has two sides. This work may also make it easier to produce fake news.

We will discuss the limitation and social impact in the revised version.

---

### Decision · Program_Chairs · 2021-09-27

**Decision:**

Accept (Poster)

**Comment:**

The paper used the Jacobean of the generator and Robust PCA (on covariance of Jacobean) to find directions for editing a region A that keep a region B in an image intact. By basically projecting the robust PCA eigenvectors of region A on the null space of robust PCA in region B. The idea is intuitive and makes sense and works in practice.

After reading the reviews and the paper. I have few suggestions to clarify the paper. Although reviewers said the paper is easy to follow, and I am familiar with works in the topic  I found the description of the method to be quite unclear and not precise.

Please include a sketch of an algorithm in the paper that makes it clear to the reader what is your method, and please also make your code available since one can not reproduce this paper in its current version.

is  your method  I)
1. The method starts by finding $z_A$ and $z_{B}$ , to produce $x_{A},x_{B}$  and compute Robust PCA of $J^{\top}J$ at $z_{A}$ and $z_{B}$ get $V_{A}, V_{B}$
2. Project $V_{A}$ on null space of $V_{B}$, get $P_{A}$
3. $G(z_{A}+ \alpha P_{A,j})$  and merge with $x_{B}$ to produce new images

or II)
1. Find $z$ to reproduce image then compute robust PCA on $JG_{A}(z)^{\top}JG_{A}(z)$ , by restricting the covariance to pixels in region A , same for region B
2. Continue with steps 2 and 3 from above using latent $z$


I am left with a question to the authors that you should clarify in the final manuscript . I may have overlooked it in appendix but I did not see it. The method works because of projection to the null space of region B or because of  the use of Robust PCA.
What if you replaced in your method Robust PCA with just PCA , I think this ablation  should be included in the paper.

I recommend a weak accept but I ask the authors to include a sketch of clear procedure of how their method is applied, open source their code, and to do the ablation robust PCA versus PCA.